# A Framework for Detecting Intentions of Criminal Acts in Social Media: A Case Study on Twitter [†]

**Ricardo Resende de Mendonça** [1,‡] **, Daniel Felix de Brito** [2,‡]**, Ferrucio de Franco Rosa** [1,2,‡]**, Júlio Cesar dos Reis** [3,‡] **and Rodrigo Bonacin** [1,2,*,‡]

1   Computer Science Master Program, UNIFACCAMP, Campo Limpo Paulista SP 13231-230, Brazil; ricardo.resende@gmail.com (R.R.d.M.); ferrucio.rosa@cti.gov.br (F.d.F.R.)
2   Information Technology Center Renato Archer, Campinas SP 13069-901, Brazil; fbrito.daniel@gmail.com
3   Institute of Computing, UNICAMP, Campinas SP 13083-970, Brazil; jreis@ic.unicamp.br
*   Correspondence: rodrigo.bonacin@cti.gov.br; Tel.: +55-19-3746-6192
†   Presented at the 16th International Conference on Information Technology-New Generations (ITNG 2019), Las Vegas, NV, USA, 1–3 April 2019.
‡   These authors contributed equally to this work.

**Abstract:** Criminals use online social networks for various activities by including communication, planning, and execution of criminal acts. They often employ ciphered posts using slang expressions, which are restricted to specific groups. Although literature shows advances in analysis of posts in natural language messages, such as hate discourses, threats, and more notably in the sentiment analysis; research enabling intention analysis of posts using slang expressions is still underexplored. We propose a framework and construct software prototypes for the selection of social network posts with criminal slang expressions and automatic classification of these posts according to illocutionary classes. The developed framework explores computational ontologies and machine learning (ML) techniques. Our defined Ontology of Criminal Expressions represents crime concepts in a formal and flexible model, and associates them with criminal slang expressions. This ontology is used for selecting suspicious posts and decipher them. In our solution, the criminal intention in written posts is automatically classified relying on learned models from existing posts. This work carries out a case study to evaluate the framework with 8,835,290 tweets. The obtained results show its viability by demonstrating the benefits in deciphering posts and the effectiveness of detecting user's intention in written criminal posts based on ML.

**Keywords:** intention detection; ontology; machine learning; crime slang expression; security; OWL

## 1. Introduction

The Web and social network services play a central role to bring individuals with common interests closer. However, such close interaction enables the occurrence of illicit events. Criminals use the Web for various activities, such as building virtual communities, mobilization, provisioning information, online training, disseminating their services or ideals, recruiting new members, and financing and mitigating risks. In practice, Web content and its communication capabilities provide advantages for planning and executing criminal acts [1]. On the other hand, the Web can benefit the development of software tools and mechanisms to support the investigation and the prevention of criminal acts.

Several countries, such as Brazil, have seen a fast increase in crime rates in the last few decades [2]. This leads to difficulties in dealing with criminal occurrences in a timely and effectively way. To this end, automated services are necessary for both the prevention and investigative processes [3]. Nevertheless, it is still necessary to research more efficient methods to analyze the criminal content in the context

of social networks [4], such as the analysis and detection of intentions related to crimes [5]. It is desirable, for example, to distinguish between who is inducing a person to commit a crime and who is commenting on a crime announced by the media. Therefore, automated analysis of users' intentions in social network posts can support investigators to understand the goals of the posts.

Automated analysis of user's intentions described in natural language posts remains an open challenge. According to Wu et al. [6], the language used in social networks is short and informal, which represents a computational challenge for automated mechanisms. The use of slang, such as those expressed by criminals, incorporates complexity in designing effective computational algorithms. In this investigation, we refer to the dialect of (cyber) criminal as Criminal Slang Expression (CSE).

The identification and classification of crimes mediated by the Web are hard tasks. There is a huge amount of Web content as unstructured data, which makes it arduous to analyze manually [7]. The complexity is even higher when people use CSE, which is not formally defined. It is used to hinder others from understanding. This produces a "secret" language, which can be used in conjunction with data encryption (out of the scope of this paper) [8]. Despite the use of complex cryptography techniques [9,10], which is not always viable, criminals use marginal/group CSE for a more restricted communication to a specific group. In addition, the volume and velocity of crimes execution committed with the support of the Web (e.g., drug trafficking and terrorism) [11] make it harder to deal with this problem.

Literature presents lexical dictionaries aiming to represent CSE. For instance, Mota [12] presented an extensive work with CSE used in Rio de Janeiro, Brazil. These dictionaries are mostly built to support human (criminal investigators) in interpreting CSE. Nevertheless, formal models suited to represent CSE that can be interpreted by computers are still needed. Such models should provide flexible and expansive constructions because CSEs are in constant evolution. According to Agarwal and Sureka [13], only lexical dictionaries are not enough for building automatic intention detection mechanisms. Automated detection using an exclusively lexical approach proved to be flawed [14], requiring the adoption of multiple techniques, such as Machine Learning (ML) ones.

Techniques and improvements in the area of categorizing emotions and feelings (that addresses intentions indirectly) have been explored (cf. Section 3). However, literature shows that evaluation and representation of criminal intentions in natural language texts are rare. In this context, there is a lack of further studies exploring the use of linguistic fundamentals of intention analysis in written texts, such as Semiotics [15] and Speech Act Theory (SAT) [16,17].

In this paper, we present a framework for selecting and classifying social network posts with CSE. We combine Semantic Web technologies and ML algorithms to this end. The Semantic Web provides computer interpretable models [6], which are useful for representing semantic relations between domain concepts. Law and security concepts [3,18–20], for instance, can be better interpreted (by human and computers) through the formal representation using ontologies. We chose to use an ontology to describe aspects related to the language used to commit criminal acts. The use of ontologies allows us to describe relationships and make inferences to determine weights related to suspicious messages, which cannot be represented by simple vocabularies, or other less structured knowledge representation systems. In addition, ontologies are expandable and interoperable models on the Web, which can be (re)used by other Web systems.

Our work proposes the Ontology-Based Framework for Criminal Intention Classification (OFCIC). The solution explores automatic classification models and algorithms applied to short textual messages to help in the detection of digital criminal acts. We propose the Ontology of Criminal Expressions (OntoCexp) [21] to provide a formal and extensible model for representing CSE on social networks. The framework OFCIC uses the ontology OntoCexp for selecting potentially crime related posts, as well as to automatically decipher the posts. The ML techniques are used for automatically classifying the posts according to an intention classification framework [22,23].

Our solution provides computational mechanisms and a software prototype to support investigators in the task of selecting potential criminal posts and filter them according to a predefined

intention classes from SAT. This work contributes and differs from other studies by proposing a hybrid framework, which uses fundamentals, concepts, and techniques from Semantic Web, Semiotics, SAT, and ML. It uniquely addresses intention classification (using SAT) of social media posts using CSE. Our solution is useful to assist human investigators in law enforcement agencies in the detection and analysis of suspicious social media posts. It allows for selecting and ranking posts from social media according to inferred suspicious level and intention.

This investigation is applied and evaluated in an empirical study on Twitter®. Based on 8,835,290 tweets analyzed, 702 tweets were filtered from this set using the framework and used as our ML training and test dataset. A prototype software tool provides functionalities for recovering potential criminal tweets (using the OntoCexp) and for filtering classified posts according to intention classes.

The remainder of this article is organized as follows: Section 2 introduces the theoretical and methodological background by including key concepts; Section 3 presents the related work; Section 4 details the OFCIC framework, including the OntoCexp; Section 5 presents our experimental evaluation with Twitter; Section 6 discusses the achieved results; finally, Section 7 concludes the paper.

## 2. Theoretical and Methodological Background

This section presents the theoretical and methodological foundations. Section 2.1 presents an introduction to social network services and discusses the spreading of crimes through them. Section 2.2 presents the technologies under this proposal by including: principles and methods in the Semantic Web, Ontologies, Resource Description Framework (RDF), RDF-Schema, and Ontology Web Language (OWL). Section 2.3 describes the theoretical background concerning Semiotic theory and methods, Speech Act Theory, and illocution classification. In addition, Section 2.4 presents the employed machine learning algorithms for intention detection.

### 2.1. Social Networks and Crimes

A Social network is understood as a set of interconnected people through a shared interest regarding friendship, kinship, sexual and corporate relationships [24]. An accurate view of this definition takes into account the relationship between the virtual representation of individuals with common interests.

The ubiquity of social networks has drastically changed the manner humans share information, resulting in new fields of research [25]. The popularity of social network services is increasing year after year. Other terms are used to represent social networks, such as: virtual communities, systems or sites of social networks in the Internet, online social networks, and social networks site [24].

Social networks provide a large amount of data for analysis, as well as research challenges such as detection of emotions and user intentions. In addition, the intrinsic anonymity of digital activities contributes to the execution of socially recriminated behaviors such as racism, homophobia, and other crimes [25].

As the benefits offered by social networks are broadly accepted, it is often convenient to accept or ignore their disadvantages. This is a matter of concern given the continuing rise in misuse of social networks [25]—for instance, postmodern terrorism benefits of technology, particularly communication technology, allowing the planning, coordination, and execution of criminal acts. These acts are committed without the presence of territorial, political, and financial barriers. Criminals maintain channels for communication using online social networks and thousands of web pages for their own interests, exploiting an unregulated, easily accessible and highly anonymous area [26].

Street gangs dominate physical territory based on public threats, violent behavior, or criminal activity. Online demarcation through social networks occurs in a similar way. The strong engagement of users in social networks has allowed criminals an advantage never before seen. Prior to the popularization of the Web, criminals' area of activity was restricted to geographical barriers. Now, much of the population is surrounded by technology, social networks, and thus by online criminal activities [27].

The Social Networks Analysis can be used to investigate or anticipate criminal activities, profiles, relationships and posts. This allows identifying relevant information about operations performed by criminals, as well as location and participants [27]. Criminals can hide their communication in a variety of ways, such as images, videos, encryption, or steganography techniques, thus making it difficult to combat crimes over the Web [28].

Crimes such as pedophilia are also practiced with the support of social networks. Pedophiles use social networks to attract their victims. The alienation occurs initially in a discreet way, aiming at the isolation of the victim. With the victim isolated, sexual predators change their approach and use direct texts with sexual content or video communication. This process has the purpose of enabling a face-to-face meeting [29].

## 2.2. Semantic Web and Ontologies

The Semantic Web has incorporated capability of understanding Web content into machines. This ability is typically related to documents and structured data, not being extended to speech or writing in natural language [30]. In the Semantic Web context, an ontology is defined as "an explicit specification of a conceptualization" [31]. The term conceptualization refers to the meaning of concepts and their relationships in a given context; the term specification is defined as the declarative, formal, and explicit representation of concepts and relationships [32]. An ontology is a formal vocabulary that usually addresses a specific domain, where definitions are formulated by describing the relationship between terms [33]. The development of an ontology requires deep knowledge of the domain to be modeled. The effort spent on development is rewarded because an ontology can be employed in many situations [34].

The relationship between four sets is used to define an ontology: (1) the set of classes is responsible for representing the concepts of a specific domain; (2) the set of relationships or associations defines the way classes are connected; (3) the set of instances formulated based on the proposed classes; and, finally, (4) the set of axioms that aim to define intrinsic constraints and rules on instances [32]. The representation of an ontology can be performed in two manners: (i) Formal, used to enable the consumption of ontologies by automated mechanisms; and (ii) Graphic, used to aid human understanding. Formal representation uses expressive description languages, such as RDF-Schema and OWL (https://www.w3.org/TR/2012/REC-owl2-overview-20121211/) [32].

There are several types of ontologies. Those ontologies independent of a specific domain are classified as "high-level" (top ontologies), thus allowing the reuse of generic concepts by more specific ontologies. Ontologies classified as "domain" specify concepts of a given domain. The classification of ontologies as "task ontology" occurs when a vocabulary related to a generic task is constituted through the specialization of the concepts present in a high-level ontology. Finally, "application ontologies" refer to functions performed by domain entities in the execution of a specific task [35].

We identified various research directions in the domain of ontology-based social network analysis. An opinion mining system based on a type-2 fuzzy ontology (T2FOBOMIE) was proposed by Ali et al. [36]. The system reformulates the user's full-text query to extract the user requirement and converts it into the format of a proper classical full-text search engine query. It retrieves the reviews and extracts opinions from these reviews by using a fuzzy domain ontology. Ali et al. [37] proposed a classification technique for feature review's identification and semantic knowledge for opinion mining based on Support Vector Machine (SVM) and Fuzzy Domain Ontology (FDO). The proposed system retrieves a collection of reviews about the hotel and its characteristics. The SVM identifies hotel feature reviews and filters out irrelevant reviews (noises); FDO was used to compute the polarity term of each feature. According to the authors, the use of FDO in conjunction with SVM increased the accuracy rates of review classification and of opinion mining. In another study, Ali et al. [38] explored an SVM and fuzzy ontology-based semantic knowledge system to systematically filter web content and to identify and block access to pornography. Their work classifies URLs into adult URLs and medical

URLs by using a blacklist of censored web pages. The proposed fuzzy ontology extracts web content to find website type (adult, normal or medical) and block pornographic content.

Ontology-based frameworks were studied for the extraction of adequate information from tweets and reviews. In [39], the use of fuzzy ontology-based sentiment analysis and rule-based decision-making was proposed, aiming to monitor transportation activities (accidents, vehicles, street conditions, traffic, etc.) and to make a city-feature polarity map for travelers. A fuzzy ontology and an intelligent system prototype were developed. The prototype retrieves the reviews and tweets related to city features and transportation activities. The opinions were extracted from the retrieved data, and a fuzzy ontology was used to determine the polarity of the transportation and the city. A merged ontology and support vector machine (SVM)-based information extraction and recommendation system was proposed in [40] aiming to suggest items with a positive polarity term to the disabled user. It mines full-text queries to extract disabled users' needs, and converts the query into a correct format for a search engine. It downloads a collection of information about items (city features, diabetes drugs, and hotel features). SVM was used to identify the relevant information on the item and removes anything irrelevant. Merged ontology-based sentiment analysis was employed to find the polarity of the item for recommendation. An ontology and latent Dirichlet allocation (OLDA)-based topic modeling and word embedding approach for sentiment classification was proposed in [41]. Transportation content was retrieved from social networks, irrelevant content was removed to extract meaningful information, and topics and features were generated from extracted data by using OLDA.

### 2.3. Semiotics, Speech Act Theory, and Illocution Classification

Semiotics is the science of the formal doctrine of signs, originating from the Greek word sémeiötiké. Semiotics encompasses the entire process of constructing a sign, from its origin to processing and effect [15]. Various areas employ the methods and theories of Semiotics, highlighting the areas of linguistics, philosophy of language, knowledge engineering, and media studies. By expressing a sign, an individual pursues three goals: (i) express a meaning; (ii) transmit an intention; and (iii) produce a social effect (e.g., convincing, obtaining, realizing). A sign can be intentionally employed for a specific purpose [42].

The SAT was initially developed by Austin [17] and Searle [16] with the purpose of enabling further communication analysis considering a communication act as the minimal unit of analysis. In this theory, propositions and intentions are present in communication between speakers (i.e., in communication acts). Propositions and intentions are not restricted to the representation of situations in a given world, but may contain a statement, a question formulation, a request, an order, a suggestion, or a manifestation of will. These characteristics demonstrate that the speakers perform various acts during the communication. These acts are classified as illocutionary acts [43].

Verbs presented in a statement are closely related to the proposed action, called a performative verb by Austin [17]. The distinction between the affirmative and performative statements is characterized by: (i) affirmative statements describe facts and events; and (ii) performative statements are formulated aiming at accomplishing something. An affirmative statement can be considered true or false, whereas a performative statement cannot be evaluated in such a way, being considered as successful or unsuccessful [44].

The distinction of speech acts on three levels was proposed by Austin [17] and Searle [16], namely: locutionary, illocutionary, and perlocutionary acts. The locutionary act refers to a sequence of words in accordance with conventions at their phonetic, syntactic, and semantic levels. The illocutionary act refers to the actions intended by the speaker at the moment the statement is produced (e.g., to warn, to thank, to advise, to ask). The perlocutionary act arises from the consequence or effect caused by the illocutionary act. In this work, we focus on illocutionary acts, as they refer to the intentions of those who perform a speech act.

The illocutionary act can be considered as the core of the speech act by performing a very important aspect: the illocutionary force. This force culminates in the performative verb, establishing

the type of the performed act. Performative verbs usually describe the illocutionary forces undertaken. However, implicit performative verbs can also be used, maintaining the illocutionary force. Thus, the determination of illocutionary force is not entirely related to linguistic elements. A classification of illocutionary forces into five classes was proposed by Austin [17], namely: verdictives, exercitives, commissives, behabitives and expositives. The illocutionary purpose (or illocutionary objective) can be considered the most important component of the illocutionary force, and it has five types, namely: Assertive, Commissive, Directive, Declarative, and Expressive [45].

Communication is applied to allow the creation, modification, or fulfillment of social commitments. According to Liu [22,23], a communication act consists of: speaker, recipient, and the message. The message can be divided into two parts: (i) function, which is responsible for specifying the illocution that characterizes the speaker's intent; and (ii) content, which represents the meaning of the message, being strongly dependent on the environment where the proposition is made [22,23].

Liu [22,23] proposed a framework for the classification of illocutions (Figure 1). This framework is grounded in the Speech Act Theory and the Semiotic Theory. In his proposal, illocutions are grouped into three dimensions: invention, mode, and time. Our investigation adopts such classification of illocutions.

Liu's conceptual cube [22,23] has three dimensions, where each dimension is divided into two groups. It allows the classification of illocutions into eight types. Each cell of the cube (Figure 1) is labeled and representative verbs are linked to illustrate each illocution class. The "Invention" dimension can be classified into descriptive and prescriptive inventions. If communication has an inventive or instructive effect, and the illocution is classified as prescriptive, otherwise descriptive. The dimension "Mode" classifies how affective (affective mode) is the expression; otherwise, it is denotative (denotative mode). Finally, the dimension "Time" is based on the social effects produced, namely: past, present, or future [22,23].

For example, the tweet "*Caveirão acabou de subir na Fazenda!!*" (in Portuguese language) (in English—"A tactical operations armored vehicle just climbed the Farm!!") can be classified as an *assertion*. This classification arises because the temporal dimension is identified in the *past*; the invention dimension is descriptive because a description is about a fact contained in the message; and the mode dimension is *denotative* because the message is not judging or accusing something.

In this investigation, we adopted Liu's Classification [22,23] as a reference because it is based on clear and representative dimensions; and its concepts allow the joint analysis of intention (illocutionary act), which represents the function and the content to which it refers. Bonacin et al. [46,47] proposed the Communication act Ontology (CacTO), which was designed to represent concepts of the pragmatic communication analysis. In this ontology, the communication acts are classified according to three dimensions (invention, time, and mode). Each communication act is linked to its performer and addressee. The message class includes Dataproperties (float values) to represent each illocution dimension and to describe the function part. The content is represented by links to external domain ontologies (using Objectproperties). Preliminary results showed the increase of the F1-Score from 0.77 (using only domain ontologies) to 0.86 (using CacTo integrated with domain ontologies) on scenarios of the special education domain [47]. A similar approach was explored on scenarios of electronic health records retrieval [48]. The F1-Score has increased from 0.65 to 0.76 by considering the illocution dimensions.

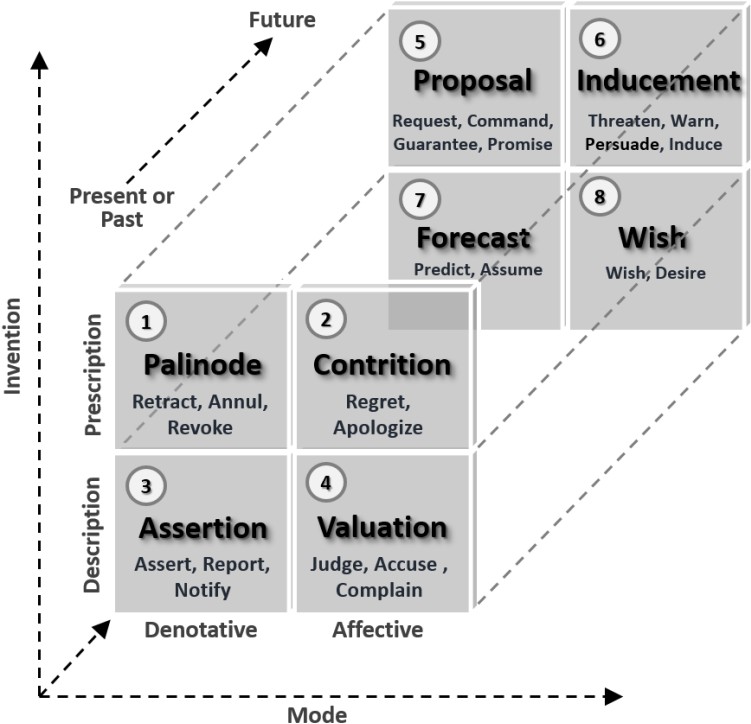

**Figure 1.** Illocution classification framework (adapted from Liu [22]).

## 2.4. Machine Learning Algorithms

We applied ML techniques for the purpose of classifying natural language posts through Liu's framework [22,23]. To this end, we introduce techniques selected from our systematic literature review (cf. Section 3). Considering the task of text classification, we explore ML techniques presenting better performance, namely: Support Vector Machine (SVM) and Neural Networks. In addition, we consider Naive Bayes (which is also frequently used for text classification), and Random Forest (which also showed good results [13]).

SVM and Neural Networks are based on optimization methods, which represent data by using the optimization of functions. SVM is based on the theory of statistical learning and establishes several principles to be followed, aiming to generate classifiers with good generalization [49]. In supervised tasks, the formulation considers the label of objects, in which data examples are annotated with their respective classes. SVM have received great attention from the ML community over the last years, with comparable results, and are often superior to established algorithms in various tasks as compared with its peers [50].

Neural Networks present several characteristics that justify their application in text classification tasks. In addition to good performance and generalization, fault and noise tolerance stand out. The decisions made by the algorithm are based on a large number of variables. In this article, we use a multilayer neural network. A traditional multilayer Artificial Neural Network is a layered model composed of several processing elements called neurons. This model presents an input layer, where data are received by the neural network, one or more intermediate layers, and an output layer, which provides the classifier response. Each neuron has a function called Activation Function.

The connection between two neurons of different layers is called synaptic weight. The input value for input layer neurons corresponds to the data to be used by the neural network. For the other neurons of other layers, the input value in the neuron, i.e., the input value of the activation function of a given neuron, is equal to the sum of the outputs of the previous layer neurons multiplied by the respective synaptic weights.

Naive Bayes classifiers are used for extensive text classification, based on word frequency assessment as characteristics [51]. This is a scalable classifier, which offers good empirical results [51,52].

Random Forest is a supervised machine learning method used for data classification and regression, based on multiple decision trees. An important aspect of Random forest is that each decision tree works independently of the others by having a random subset of characteristics. The final decision of the forest is based on the vote of each tree.

Details about the Neural Network, SVM, and Random Forest explored in this work are presented in Section 5.1.1. Further details about Neural Network and SVM theory is described in [49]. Additional details about Random Forest is described in [53].

## 3. Related Work

This section presents a literature review, which aims at answering the following research question: "What are the approaches to computationally assess and represent the intent of criminals in posts written in natural language using slang?". The review was methodologically based on Kitchenham's guide [54].

In our methodology, a preliminary and exploratory investigation based on the research question was carried out aiming at obtaining the necessary inputs for the literature review, resulting in: (i) research parameters; (ii) scope of the search; (iii) scientific search databases; (iv) keywords; (v) search fields (e.g., title, abstract, manuscript). As we expected to identify advances and contributions in recent years, the search period was considered from 2014 to 2019 considering the fact that the research question addresses relatively recent research issues. We used the following search string: "*(ontology OR thesaurus OR taxonomy OR vocabulary) AND (intention OR semiotics OR "speech acts") AND (crime OR criminal)*". The syntax was adapted to each scientific search database.

We considered all returned articles from all databases, except the Google Scholar. Due to the scope of this database (indexing of other databases), the first 100 articles were considered (The google scholar returned thousands of results in a relevance order. As no relevant titles were found between the 100th and 200th results, we decided to stop the evaluation in the 100th paper). The initial search obtained a total of 1876 articles; 289 articles from Springer Link, 346 from IEEE Xplore, 763 from Science Direct, 378 from ACM Digital Library, and 100 from Google Scholar.

The inclusion and exclusion criteria were defined by three of the authors in an iterative process of reading articles (in the exploratory search) and proposing criteria until reaching consensus. We detail the Inclusion (I) and Exclusion (E) criteria as follows:

- I1—Research on analysis of intentions in natural language over social networks;
- I2—Research on analysis of encrypted language (e.g., slang);
- I3—Studies that use the Speech Act theory or Semiotics for analysis of encrypted languages;
- I4—Studies exploring ontologies to represent knowledge about criminal acts;
- E1—Articles written in languages other than English and Portuguese;
- E2—Articles that are not related to intention or emotion analysis in addition to at least one of the following themes: ciphered language analysis, speech acts theory, ontologies, and semiotics;
- E3—Articles that are not related to computing or multidisciplinary with computing;
- E4—Texts other than scientific publications;
- E5—Summary papers with less than four pages that do not have the depth or relevant results; and
- E6—Systematic Reviews and Books (Paper collection books had their articles evaluated individually).

Twenty-four articles were excluded due to duplicate results from the scientific databases. The remaining articles were submitted to the inclusion and exclusion criteria. The first evaluation considered the title, abstract, and keywords. Forty-two articles were categorized as papers with the possibility of adherence to the research theme, and they were fully evaluated according to the criteria. Three researchers analyzed the selected articles and prepared the final list by consensus after discussion. During this evaluation, eight studies were identified as non-adherent and six papers were classified as literature reviews related to the theme.

In the following, we present a synthetic analysis of results concerning the 27 selected studies organized by two main categories. Section 3.1 describes solutions based on Lexicon Dictionary and Ontology for intent analysis. Section 3.2 presents ML-based solutions. Section 3.3 discusses the key findings on the related work.

### 3.1. Lexicon Dictionary and Ontology-Based Solutions

Lexical dictionaries and their limitations in feelings evaluation were addressed in [14,55,56]. In Teodorescu et al. [55], each slang expression is manually annotated with meanings assigned to it. According to the authors, slang can contain numerous meanings, or even its use may be completely different in other cultures. Several sentiment analysis proposals do not adequately consider the influence of social, geographical, and cultural issues; such a factor impairs its effectiveness for intent classification from natural language posts [57].

According to Teh et al. [14], the use of a lexical dictionary is not sufficient to detect the presence of hate speech in texts written in natural language. The authors indicated that the evolution of vocabulary must be considered for this purpose [14]. Teh et al. [14] organized hate terms into eight categories. A lexical dictionary that correlates a word with a feeling from the Plutchik's wheel of emotions was used in [56]. In their work, after establishing this correlation, hashtags were evaluated to identify which event the post refers to.

Hagen et al. [58] explored ontologies, linguistic techniques, and emoticon assessment to classify emotion at the time of collecting records. In their work, an ontology represents attacks, defenses, attacker, and attacker's objective. The outcome of this phase was evaluated by a sentiment analyzer that determines whether the content is harmless or a threat. According to the authors, for static domains, a well-defined ontology is sufficient. However, for dynamic environments, constant updates are necessary, with possible changes or removal of concepts.

### 3.2. Machine Learning and Mixed Learning based Solutions

Among the 27 evaluated studies, 22 use ML techniques as a core element in the proposals aiming to identify intentions and emotions (e.g., hate and depression) in texts written in natural language. The main ML techniques identified were Neural Networks and SVM. Some studies were considered mixed as they made use of ontologies with machine learning. We present a summary of the found studies.

Garcia-Diaz et al. [51] defined the process of emotion extraction as "opinion mining", which consists of using natural language processing and computational linguistics. The data mining process takes place in five steps, namely: selection, preprocessing, transformation, data mining, and interpretation. In preprocessing of tweets, user names and user quotes were converted to an identifier to ensure anonymity. The fourth step is responsible for generating the output that should be worked out using Naive Bayes. According to the authors, the accuracy of the ML sentiment analysis classifiers was improved by using expert domain-specific knowledge from social networks.

A process of detecting depression in natural language messages was proposed in Losada et al. [59]. The study consisted of four steps: (1) selection of the text origin; (2) data extraction; (3) prediction, which consists in analyzing the message history of a given user; and (4) text classification by using a Logistic Regression algorithm.

A categorization of misogyny types was proposed by Anzovino et al. [60]. The main objective was to detect misogyny in messages posted on Twitter, by using ML and NLP for classifying tweets as misogynistic. In Waseem et al. [57], ML techniques were used with multiple intermediate processing tasks to improve assertiveness. Similar to the results obtained by Garcia-Diaz et al. [51], this process presented improvement of results but loss of time performance. Agarwal and Sureka [13] proposed a lexical dictionary to assist the detection of racism in posts on Tumblr. A linguistic analysis was performed by using two APIs (Alchemy Document Sentiment and IBM Watson Tone Analyzer). Five categories of sentiment were proposed: joy, fear, sadness, anger, and disgust. The authors

experimented the techniques: Random Forest, Decision Tree, and Naive Bayes. The results showed the superiority of the Random Forest technique for this task. In a similar task related to hate speech detection, Zhang et al. [50] explored a deep neural network based method by combining Convolutional Neural Networks (CNN) and Gated Recurrent Networks (GRN). In their evaluation, such proposition performed superior to SVM, SVM+, and CNN.

The recognition of sarcasm is a hard task, due to lack of context, use of slang, and rhetorical questions. In Ghosh et al. [61], the detection of sarcasm in social networks and discussion forums was investigated by using SVM and Long Short-Term Memory (LSTM). In their study, LSTM obtained better results than SVM.

Classification techniques were used by Justo et al. [62] (rule-based classification and Naive Bayes) in the identification of sarcasm and malice in comments posted in social networks. Although initially seeming similar, sarcasm and malice are distinct in the form of detection. For sarcasm detection, it is necessary to know the domain and include characteristics (e.g., context) for better evaluation. In their study, other techniques for sarcasm detection were added, namely: lexical categories using n-Grams and sentiment detection through semantics.

Appling et al. [63] studied the use of linguistic techniques and discriminatory model to detect fraud in texts. One of the linguistic techniques used was the amount of words used in a given text to affirm something. A greater than average amount of that user indicates the possibility of fraud. This technique relies on the prior knowledge of the posts made by the user. In a similar task related to fraud, Hu and Wang [52] proposed a mathematical model for using with Naive Bayes. Their comparative study showed the superiority of Naive Bayes when compared to a Decision Tree algorithm.

In Dhouioui and Akaichi [29], text mining was used to classify conversations by means of lexical dictionaries and extraction of behavioral characteristics. They proposed a combination of techniques for enhancing the detection of sexual predators. A comparison was conducted to analyze SVM and Naive Bayes. In this context, SVM demonstrated better performance. Three lexical dictionaries were developed, namely: (i) Emoticons; (ii) English language contractions; and (iii) Terms commonly used for communication via SMS.

Semantic Role Labeling (SRL) was used in Barreira et al. [64] for forensic analysis of text messages extracted from mobile devices. According to the authors, the combination of linguistic techniques and ML achieved better accuracy than when these techniques were used separately. A Random Forest and Bagging based technique was proposed by Levitan et al. [65] for the detection of fraud in video speeches. The audio transcription was required for application of the proposed technique. According to the authors, the adoption of a lexical dictionary might improve the outcome.

An approach to cyberbullying detection by means of ML techniques was proposed by Raisi et al. [66]. After selecting a pair of users, tweets were scanned to identify bullying. This approach was restricted to environments where the access to the messages exchanged between those involved was allowed.

Pandey et al. [67] developed an approach for categorization of intentions. Their approach was based on a linear model of logistic regression and CNN in addition to being supported by distributional semantic. Three categories of intentions were formulated: accusation, confirmation, and sensationalism.

Most investigations aim to detect events that have already occurred [68]. Escalante et al. [68] proposed an approach to detect intent to commit fraud and aggression on social network posts. During the detection process, an analysis of documents which were categorized by ML algorithms is performed. Profiles and subprofiles were used to support the detection process. The profile is defined by the same two dimension vectors containing words (3-grams) and their classifications. The adoption of subprofiles allows the evaluation of multiple classes, as it considers the diversity of domains.

Mundra et al. [69] studied embedding learning as a technique based on neural networks for classification of idiomatic expressions in microblogs (e.g., Twitter). In addition, the Latent Dirichlet

Allocation and Collaborative representation classifiers were evaluated [5]. According to the authors, a crime intention detection system should combine more than one ML technique.

Characteristics of sounds emitted in voice communication were explored for detecting mood and user dialect [70]. Fourier parameters support detection and later categorization of mood and dialect; SVM was used in categorization. Recurrent Neural Network (RNN) and Distributed Time Delay Neural Network (DTDNN) were further evaluated, but in neither scenario were they superior to SVM. A predictive model was proposed by Aghababaei and Makrehchi [71] for predicting crime in a given area by analyzing posts from Twitter. The model combined public FBI data in some states of the United States with tweets of the most active users and the older registered accounts. Due to the lack of data in supervised learning, the authors proposed to automatically annotate tweets with labels inferred from crime trends. These annotations were the basis to automatically create training examples to be used by prediction models.

ML techniques were used in Park et al. [72] to create an ontology. According to the authors, semantic knowledge acquired by means of ontology allows a better understanding and categorization of phishing attacks (attempt to obtain personally identifiable information). Another study aimed at understanding the difference in behavior and users' intention when compared to socialbots on Twitter [73]. They extracted bag-of-words feature sets represented as TF-IDF (Term Frequency-Inverse Document Frequency). These sets were used to compare bot and human tweets. Then, they applied sentiment analysis using the LabMT tool to better understand the nature of words used by bots and humans.

### 3.3. Discussion on Related Work

We identified techniques aimed at categorizing emotions that address intentions indirectly. Studies addressing further evaluation and representation of users' intentions in texts written in natural language are very scarce. Most of the analyzed investigations use machine learning to evaluate sentiment. Natural language processing techniques are explored in the approaches proposed in [52,58,60,62–64,74,75]; data mining and semantic analysis are used in [51]. In general, the studies presented better assertiveness measures in the categorization of sentiment using complementary approaches (e.g., semantic descriptors with ML) as compared with a single one. Although this represents a decrease of response time, the benefits of adopting complementary techniques may outweigh the computational cost.

The probabilistic classifiers are reliable, but they strongly rely on the size of the training base. Some of these classifiers are not suitable for real-time data analysis, making preprocessing an indispensable task for better accuracy. This preprocessing usually demands a high computational cost [51]. The approach proposed by Garcia-Diaz et al. [51] revealed gain in performing postprocessing where improved classifier accuracy was achieved by applying domain-specific knowledge.

The study presented by Anzovino et al. [60] obtained better results by using Token N-Grams in combination with SVM. In [63], the authors point out the benefits of adopting linguistics features in combination with ML techniques.

The adoption of multiple techniques for detecting intentions and sentiment proved to be a promising practice. The use of semantics (e.g., ontologies) in conjunction with these techniques is still an open research area of study, which requires continuous efforts for obtain advancements.

The exclusive use of a lexical dictionary presented drawbacks in [72]. This characteristic was not present when using ontologies for semantic analysis of texts written in natural language. The use of NLP techniques used in conjunction with ontologies demonstrated good results in [74], reaching an accuracy of 86%.

Although emotion and sentiment are strongly related to intentions, studies focusing specifically on intentions are rare, particularly those with consistent theoretical underpinnings about the understanding of intentions. For example, only the study presented by Hu and Wang [52] explicitly

used the speech act theory, despite the importance of this theory to the problem of meaning in linguistics and philosophy.

The perspective in which words are used to accomplish things has much to contribute, not only in the detection process, especially in establishing what one wants to detect. For example, illocutions (acts of speech or writing that constitute actions) may result in different pragmatic effects, depending on the interpretation of the speaker's intentions. Thus, an illocution classification model can contribute to the definition of what we want to detect regarding "user intentions", such as committing crimes.

We also analyzed the linguistic foundations used in our related work, in particular semiotics and pragmatics studies. Although Semiotics was a search keyword, no studies analyzing posts related to intentions to commit crimes were obtained. Semiotics provides us a vast theoretical and methodological basis for understanding the use and interpretation of signs in computer systems [76]. From the semiotics' point of view, people communicate by sharing signs through multiple media.

The "Pragmatics" [77] is a field of research that aims to understand the relationship between signs and people (Pragmatics can be understood as a subfield of semiotics (and linguistics), and it focuses on understanding the relationship of context and meaning, including intentions), being essential to identify the speaker's intention. From this perspective, it is important to understand how signs influenced the communication process to identify the expression of criminal intent in social media posts.

Regarding the investigations on the realm of ontology-based social network analysis, a fuzzy ontology (T2FOBOMIE) was proposed by Ali et al. [36] to support an opinion mining system, whereas our ontology (OntoCexp) supports the identification of criminal expressions in the ciphered language (informal posts and using slang) used by criminals. Ali et al. [37] focused on the classification of reviews about a hotel and their characteristics. In our work, we aim to identify the intent to commit crimes in Twitter's posts. Ali et al. [38] aimed to filter web content and to identify and block access to pornography. They were not concerned with the use of slang, which is recurrent in pornography. In our approach, we aim to predict crimes, by looking for posts in the Twitter that contain expressions related to weapons, violence, cyber attacks, terrorism, etc.

Regarding ontology-based frameworks, in [39], the authors used sentiment analysis in opinions to monitor transportation activities (accidents, vehicles, street conditions, traffic, etc.), whereas our focus is to detect specific intentions in social media. A recommendation system was proposed in [40] aiming to suggest items with a polarity (positive or negative) to disabled users; we aim to identify criminals in social media to anticipate and avoid crimes. An approach for sentiment classification was proposed in [41] to extract meaningful information on the domain of Transportation. Differently, our goal is to extract ciphered messages from criminals to support the detection of malicious acts. In [39–41], the authors do not present solutions for slang assessment, nor take into account the use of sarcasm, which is very common in publications of social networks.

In this sense, we observe a lack of theories, methods, and tools for analyzing, representing and detecting criminal intention in social media posts—for example, studies on the interaction of users (human) with technological artifacts; analysis of linguistic aspects and their relationship to user behavior; and the application of advanced ML techniques.

Our proposed solution further advances the state-of-the-art in the automatic detection of criminal intention in social media natural language messages. Our ontology-based solution is suited to represent the target slang language to support the identification process in distinct tasks. We combine it with speech acts theory in the definition and implementation of ML-based mechanism for intention recognition in texts relying on illocution classes.

## 4. Ontology-Based Framework for Criminal Intention Classification (OFCIC)

We propose the Ontology-Based Framework for Criminal Intention Classification (OFCIC) aiming at detecting intention of commit criminal acts. Section 4.1 provides details of the framework. Section 4.2 presents the Ontology of Criminal Expressions (OntoCexp) used in the execution of the framework.

### 4.1. Framework Description

The OFCIC framework aims at providing data that supports selection of suspicious posts and intention classification, i.e., providing more accurate selection regarding a particular post using CSE. The proposed process results in availability of classified data for analysis or message selection by agent investigators. Figure 2 presents an overview of the OFCIC framework. It is compounded by two main components: the component for building the trained ML model (A in Figure 2) based on input posts; and the component to select new suspicious posts and intention classification (B in Figure 2).

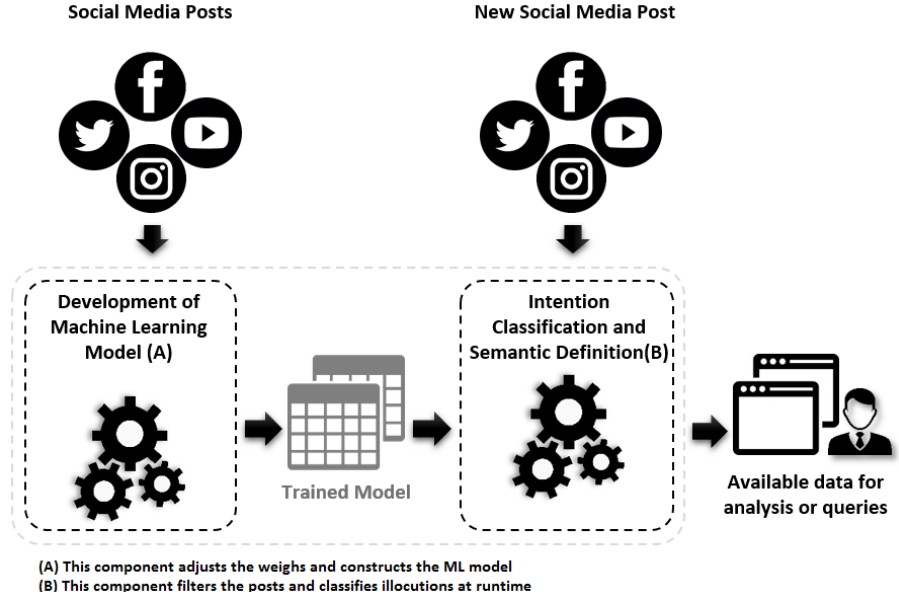

**Figure 2.** Overview of the OFCIC.

Each post available in OFCIC consists of two parts: function and content [22,23]. Liu's framework is used to classify messages' function part, specifying the illocution that characterizes the speaker's intent (i.e., we classify the function as one of the six illocution classes). The OntoCexp is used to classify the terms used in the content part, which represents the meaning of the message. We first generated a training and test set by manually assigning post functions to one of the illocution classes. Once trained, ML techniques are used to automatically assign an illocution class to posts.

During the training phase, terms of the content part are associated with concepts of the OntoCexp. Each term (individuals) belongs to a class (which defines a concept). Each term has a dataproperty defined as a weight value, which represents the chance of a term being related to CSEs. These weights are manually and interactively defined by the training component (A in Figure 2), and they are used to select suspicious posts (B in Figure 2). For example: when processed by OFCIC, the post "*Caveirão subindo a favela, prepare*", which can be understood as "A Tactical Operations Armored Vehicle is Climbing the Slum's streets, be prepare" (in the English language—translated by the authors), is classified as an inducement regarding its function; its content is identified as a confrontation with the police (represented as a criminal act). This criminal act has a weight, which is further used to select suspicious posts.

#### 4.1.1. OFCIC Training Component

Figure 3 presents the main steps for building the trained ML model (A in Figure 2).

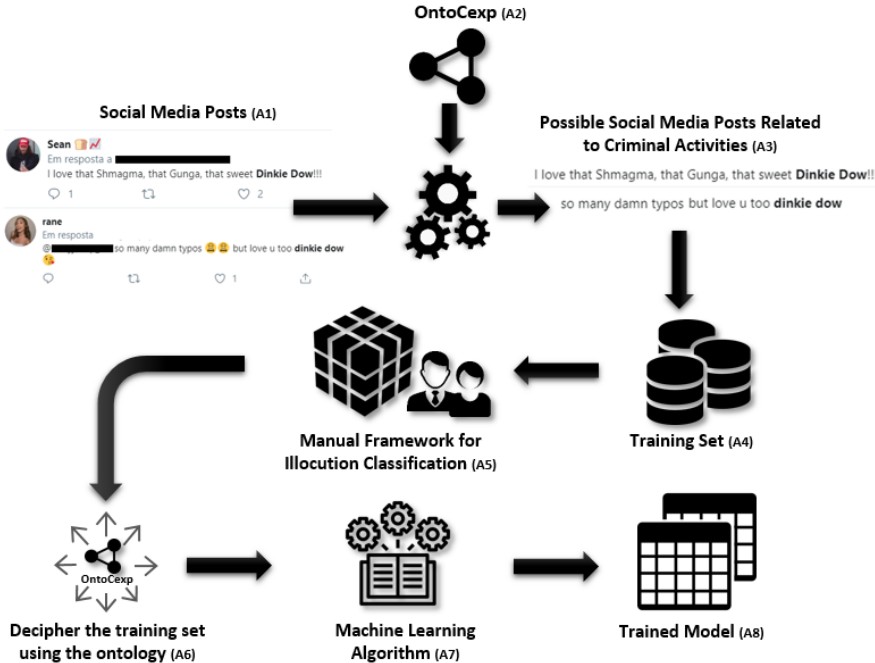

**Figure 3.** Machine learning training model in the the OFCIC.

Input data are small text posts extracted from social networks. It is necessary to filter those posts potentially related to crimes. The amount of posts extracted from social networks (A1 in Figure 3) is so large that manual manipulation for the filtering becomes unfeasible.

The classification of posts as candidates for association with criminal activities (A3 in Figure 3) is based on the concepts defined in the OntCexp (A2) ontology [21] , which was created specifically for this purpose (cf. Section 4.2). The candidate posts are selected based on instances presented in the ontology. These instances were formulated based on the terms of the glossary proposed by Mota et al. [12].

The selection of posts needs to consider those terms more related to crimes. However, various terms of the glossary are used in the daily communication of the population, which is not involved in criminal activities. To deal with this characteristic, we assign weights for the terms. Generic terms can be found in thousands of non-criminal posts. Each term in the ontology has a weight depending on its relevance for the criminal domain. More generic terms (e.g., medicine, salty, frog) were manually assigned weight *initially 1*, and more crime-specific terms (e.g., shaking the prison, biting the rope, sitting on the bamboo, suffocation) were assigned weight, *initially 3*. Similarly, suspicious acts that are defined by Semantic Web Rule Language (SWRL) [78] (or axioms) have associated weights (cf. Section 4.2). In this component, the weights (i.e., chance of a term being related to CSE) are iteratively adjusted. A researcher (or domain specialist) defines weights, uses them to select posts, and adjusts them according to the relevance of the selected posts. In our case study (cf. Section 5), a researcher (one of the authors) proposed adjustments and other two researchers validated it.

Once with adjusted values, an input post can be classified as "more relevant" to our domain of interest depending on the sum of all CSE, or combination of CSEs defined using ontology rules, which define criminal acts (cf. Section 4.2). For each post, we process automatically the weights sum. The posts are ranked and included in a training set (A4 in Figure 3). Our defined ontology plays a key role in the selection of suspicious posts from the social network and for the adequate selection of data for building the training dataset.

In order to build a labeled training base to be employed in ML algorithms (A7 in Figure 3) and create the training model (A8 in Figure 3), in this step, we manually assigned illocution types (A5 in Figure 3) for the posts included in our training set (A4 in Figure 3). It is desirable that the assigned illocution types be checked by other researchers (or domain specialist). For this purpose, we rely on

the illocution classification framework proposed by Liu [22,23] for posts classification. This conceptual framework is more suited to this study than other models proposed in [16,17]. For the models originally defined by Austin [17] and Searle [16], the act of speaking is not correlated with the object or with the content of the statement.

In step 6 (A6 in Figure 3), CSE are translated to standard language (Portuguese in our case). An application for direct translation (*i.e*, CSE to standard language expressions) is available in GitHub (https://github.com/ricardoresende/SlangWordsTranslation). This application uses the hierarchical structure of the ontology to identify generic and specific expressions. The ontology hierarchy is first included in the database (for performance reasons) and the software application reads the CSE by translating each CSE in sequence. The post "*Trás cimento para eu dar um pico*" (Bring cement for me to peak—in a word-to-word translation to English language), for instance, is deciphered to "*Trás cocaina para eu consumir droga*" (Bring cocaine to me to take drugs—in the English language—translated by the authors). In the first CSE, a more specific class (*Cocaine*) is used (because the CSE is associated with it), in the second one, a more generic class (*Drug*) is used in the translation. They are both associated with drug consumption property. Although more complex (and efficient) translation algorithms can be used in this step, it is beyond the scope of this article to define such algorithms. Deciphering the sentences with CSE is relevant to provide further data for the training phase (A7 in Figure 3).

In step 7 (A7 in Figure 3), the framework applies ML algorithms for building a multiclass classification model. In this study, the illocution classes are the output classes in the classification model. To this end, in our case study, we investigated the use and the comparison of four ML algorithms: Neural Networks, SVM, Random Forest, and Naive Bayes (described in Section 2.4). Our systematic review (Section 3), which is focused on ontologies, intentions, and crime, shows several investigations using ML, of which the most frequent ones are those mentioned above. Section 5.1.1 presents details of the setup and parameters explored in the ML algorithms in the evaluation of our study.

Finally, in step 8 (A8 in Figure 3), a classification model is generated for use in the second component of the OFCIC framework (B in Figure 2).

### 4.1.2. OFCIC Intention Classification Component

Figure 4 details the component for selecting suspicious posts and classifying intentions (B in Figure 2) for new inputs with posts from social networks. Whereas the previous component has manual steps (adjustments of the ontology weight values during step A2 and manual classification of the training set during step A5), this intention classification component is automated. It uses already adjusted ontology weights and trained ML models to select posts and classify intentions.

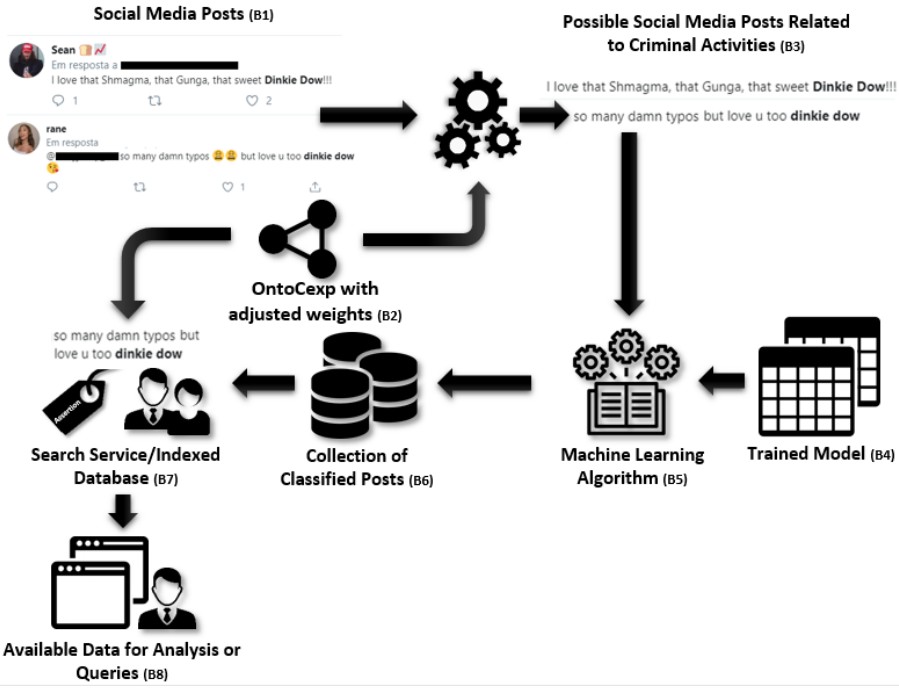

**Figure 4.** Key steps in the phrase selection and classification.

Firstly, new social media posts are collected from social network (B1 in Figure 4). Different configurations and applications should be used according to the social network service and objectives, for instance. In this work, we use the source-code in [79] to collect tweets.

The collected posts are then selected according to the CSE of each posts (B3 in Figure 4)). A suspicious level (weights sum) is attributed to each post using the OntoCexp with the weights already adjusted by the first component.

Once selected, posts (B3 in Figure 4) are submitted to ML techniques (B5 in Figure 4), which explores trained models by the first component. These techniques are used to predict an illocution class from the post according to the Liu's framework [22,23].

At this stage, classified posts are loaded and indexed in an external search service/application (B7 in Figure 4) (e.g., Elasticsearch (https://www.elastic.co/) or Solr (https://lucene.apache.org/solr/)). According to the adopted search service, ontologies can be used to improve the results. Authors in [48] describe how search mechanisms can be improved by using intention and query expansion techniques based on ontologies.

Finally, an query and analysis interface is provided (B8 in Figure 4). This may include search interfaces, reports, and visualization techniques. Section 5 presents a search interface prototype whose results (posts) are presented in a rank according to the suspicious level and intention class. The suspected posts are available for data analysis. OFCIC has a graphical user interface to assist domain experts in this process. It includes a query interface where users can choose keywords, which are submitted to search services, and filter options to select the posts according to the illocutions classes (B8 in Figure 4). Data analyses and visualization tools can be further explored using the generated data. This step uses data already collected, classified, and indexed in search services.

### 4.2. OntoCexp—Ontology of Criminal Expressions

In summary, the OFCIC uses the OntoCexp in both components as follows: (1) Training Component (Figure 3) uses the OntoCexp to select (according to the use and combinations of CSE) suspicious posts to train and test the ML algorithms. The weights of CSE and acts are manually adjusted, as mentioned before. This ontology is further used to translate the posts; and (2) the Intention Classification Component (Figure 4) uses the ontology to select suspicious posts (this is an automatic

process, once the weights have already been adjusted) and to translate the messages by then submitting them for classification. This section presents a synthesis of the OntoCexp.

Section 4.2.1 describes the adopted engineering process. Section 4.2.2 presents the core model of the ontology and details key concepts. Section 4.2.3 presents illustrative (validation) scenarios and examples of rules modeled in the ontology.

4.2.1. Ontology Engineering Process

The ontology engineering process is based on Noy and Mcguinness [80], and is compounded by three iterative steps, where the last step provides feedback (e.g., missing concepts and structure changes) for re-executing the previous two steps:

1.  **Scope and reuse**. We defined the crime domain as the scope of the ontology, and performed the correlation between expressions and terms used by people in criminal activities. For reuse purposes, we evaluated the top-level ontologies SUMO, UFO-B, and DOLCE, as proposed in the studies [81–83]. However, we chose not to use them due to the particularities of the crime domain. The ontology requirements were defined by the researchers and are focused on the objective of representing CSE and associating them to standard language to translate encrypted messages. The ontology provides alternatives for representing weights and rules that determine the degree of suspicion. The solution uses it as an application ontology for selecting suspicious messages. Therefore, it is out of the scope of this ontology to represent the entire crime domain.

2.  **Ontology elements specification**. At this stage, we enumerate terms, define classes, properties, and constraints. Our proposal of conceptualization was inspired in a crime vocabulary [12], which is a result of eight years of gathering and translating criminal's terms (words or statements) in the Rio de Janeiro state, Brazil. It contains around 1000 Criminal Slang Expressions (CSE). Despite the relevance of the vocabulary for police forces, a formal model computationally interpretable such as an ontology is needed. We performed the definition of classes by using a top-down strategy. The terms and expressions were analyzed by three researchers (co-authors in this article), who made the design decisions. Various expressions (e.g., "*X9*" or "*Xisnovear*") are very different from their meanings (semantics) in the Portuguese language. For instance, the criminal slang expression "*Vomitar/vomit*" (Portuguese/English) means betraying someone by unveiling critical information. For the elucidation of the ontology elements, we analyzed a dataset (from Twitter®) related to criminal areas to understand how criminals use such slang expressions. We filtered the tweets by using terms from [12] and collected almost 5.9 million tweets by using our list of 1009 CSE. For each invocation in the Twitter API, we retrieved at most 100 tweets posted in the last seven days (first week of October 2018) of the day the search was occurring. The tweets should contain at least one CSE, due to a restriction of the Twitter API (standard). For each CSE, we collected all tweets we could get until reaching the 7-day limit. Our search was restricted to the Portuguese language to get more trustful data. Our procedure also filtered out retweets to avoid repetitions. The developed source-code used in this task is available in [79].

3.  **Generation of instances and validation**. From the statements extracted from the tweets dataset, a set of instances was generated to validate the proposal. These statements refer to terms and relationships of our ontology (OntoCexp). The collected tweets were ranked according to the number of CSE expressions, and 274 posts were used to specify representative cases (presented in the next subsection). In the validation procedure, we considered: (i) *Positive* when criminal communications used the modeled terms; (ii) *False Positive* when usual communications used the modeled terms; and (iii) *False Negative* when criminals communications did not use the modeled terms. The objective is to provide feedback for the modelers about how the terms are used in practice. We used the false negative assignments to review OntoCexp, i.e., we modeled new terms (or concepts if necessary) in this case. Various iterations were performed by the researchers until obtaining the generation of the current stable version of OntoCexp (V3). It must be continually refined, as the criminal language is very dynamic and regional. OntoCexp is publicly available in

GitHub and WebProtégé [84,85]. Section 4.2.3 presents details of the results in the evaluation of our ontology. In addition, in the second iteration, the case study (Section 5) provided feedback on the suitability of the ontology to the requirements defined in the first step.

As the objective of OntoCexp is to represent CSE, and not the entire crime domain, we made some design decisions focused on our goals. In our ontology, classes are concepts represented by CSE, while instances are terms (CSE) used to refer to these concepts. Each instance (CSE) has labels associated with variations in their writing form. Weights values are linked to these terms and refer to the degree of suspicion. All terms of crime vocabulary proposed in [12] were included in the ontology.

### 4.2.2. The Core Model of the OntoCexp

The OntoCexp is a conceptual model, and it is used to represent criminal slang expressions. The core model of OntoCexp contains key concepts of the crime domain. This model allows us to structure the domain and to understand other related terms. Figure 5 presents the key classes of OntoCexp. We structured the textual description of the concepts, and we call "groups", i.e., we grouped terms considered similar or that can be related and described in the same group and paragraph in the paper. The conceptualization of the core classes was divided into five groups of concepts, namely: (i) People; (ii) Criminal activities; (iii) Transport and communication; (iv) Entertainment and subsistence; and (v) Criminal operations. In the following, we describe the conceptual groups.

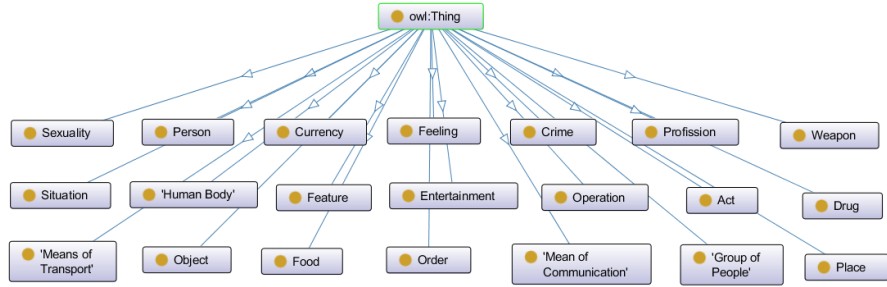

**Figure 5.** Overview of the main OntoCexp classes (adapted from [21]).

*Person*. The *Person* class refers to men and women, as well as specific aspects regarding their sexuality, occupation, etc. For instance, OntoCexp enables to represent a person, a place, a physical aggression, etc. There are several expressions regarding sexual preferences. These expressions are usual in the communication among criminals. We use the *Sexuality* class to represent this context.

*Criminal activities*. Criminal activities (e.g., obtain money, weapon, drug, etc.) were also defined, including building a cyber weapon or to launch a cyber attack. Crimes are classified as verbal (including digital) and physical. Murders and beatings are examples of physical crimes. The criminal act is related to the type of weapon used. The *Weapon* class was organized into subclasses (e.g., cold and firearms) due to their impacts in the crime measure. Criminal acts can be, for example, an armed robbery, or a cyber attack. Criminal acts are also connected to money, such as kidnapping or subornation.

*Transport and communication*. Police forces could intercept or predict the criminal acts when the transportation and communications means are identified. For instance, in a criminal investigation, identifying the vehicle used by the criminals is a key aspect due to the vehicle may have been negotiated by means of a social network.

*Entertainment and subsistence*. Criminals usually take part in illegal bet or gambling; consumption of drugs and alcohol is also usual. Classes to formalize these acts are needed because they are usually related to other criminal activities.

*Criminal operations*. *Place* refers to the geographic area where some artifacts related to a crime can be found or identified. Criminals have several manners of referring to the areas where they act. Frequently, it is critical to identify what happened (or will happen) in a specific place. For instance,

the expression *"Os coloniais vão berimbolar a gaiola"* does not make any sense in the Portuguese language, but we can extract three key elements by using OntoCexp: *"coloniais"*, *"berimbolar"* and *"gaiola"* (in Portuguese), which respectively mean "Prisoners of the Cândido Mendes Penitentiary", "Rebellion", and "Prison". From this post, it is possible to interpret the prisoners' intention in starting a revolt (rebellion) in the Cândido Mendes Prison. Understand that the criminal or police operations are also needed; ambush, escape, or invasion are examples of operations. OntoCexp is suited to represent some objects used by criminals. For instance, in the crime slang of Rio de Janeiro, the term *"Balancinho"* (in Portuguese) (child swing, in English) refers to a string object that is used by the prisoners to climb the walls of the jail to see through the iron bars windows.

### 4.2.3. Scenarios and Rule Specification

The collected tweets were ranked according to the number of CSE expressions (cf. Section 4.2.1). A researcher (first author) analyzed the tweets (from the top of the rank) and selected 274 posts. Finally, the researchers (three of authors) selected 14 representative cases to be used in the OntoCexp as validation scenarios (i.e., a preliminary evaluation focused on the capability of the ontology to express these scenarios). In this section, we describe two illustrative scenarios (cf. [21] for other scenarios) to exemplify the evaluation procedure conducted.

The post *"vamos explodir o caveirão e capota o Águia"* (in Portuguese) ("let's explode the big skull and hood the eagle"—in a word-to-word translation to English language) ("let's blow up the police car and police helicopter"—translated by the authors) refers to a proposal (illocution class) and contains two terms appearing in the Mota' vocabulary [12]. The terms *"caveirão"* ("big skull" in the English language) and *"Águia"* (eagle in the English language) refer to a police armored vehicle and a police helicopter. Figure 6 presents this scenario instantiated in the OntoCexp. Note that *"Águia"* and *"caveirão"* are instances of the classes *"PoliceHelicopter"* and *"PoliceCar"*, respectively.

Rules defined in our ontology are used to rank messages about criminal acts from collected posts of social networks. The SWRL [78] is used to express knowledge-level inferences and queries. Listing 1 presents examples of SWRL rules defined to relate the acts described in the illustrative scenario to criminal activities.

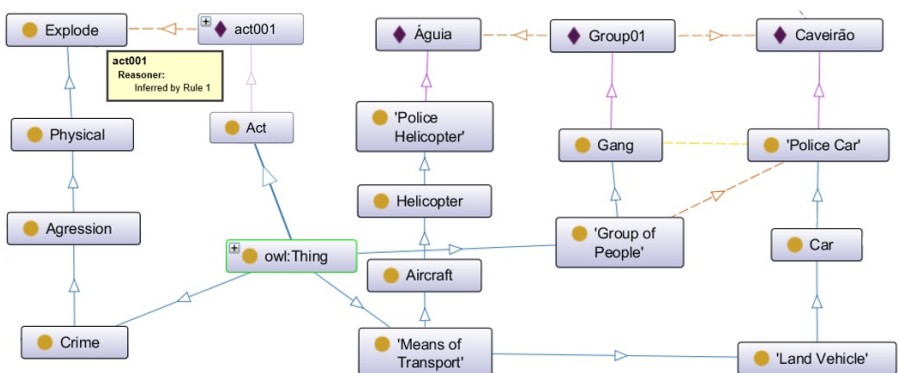

**Figure 6.** Partial representation of the Tweet (example) instantiated in the OntoCexp.

**Listing 1.** Examples of inference rules related to criminal acts specified in the ontology.

---

**Rule 1 -** Gang(?g01), Act(?ac01), action(?g01, ?ac01), 'Police Helicopter'(?h01), requestExplode(?ac01, ?h01) -> Explode(?ac01), hasLabel(?ac01, "blowUpHelicopter"), hasWeight(?ac01, "8" $\hat{x}$sd:int)
**Rule 2 -** Gang(?g01), Act(?ac01), action(?g01, ?ac01), 'Police Car'(?c01), requestExplode(?ac01, ?c01) -> Explode(?ac01), hasLabel(?ac01, "blowUpPoliceCar"), hasWeight(?ac01, "6" $\hat{x}$sd:int)
**Rule 3 -** Person(?p01), Act(?ac01), action(?p01, ?ac01), Cocaine(coc01), 'Drug use'(?du01), actUseDrug(?ac01, ?du01), consume(?ac01, ?coc01) -> 'Drug Abuse'(?ac01), hasLabel(?ac01, "cocaineConsumption"), hasWeight(?ac01, "4" $\hat{x}$sd:int)

---

During the post selection steps, each CSE receives a value according to (iteratively) predefined weights. In addition, rules in the ontology are executed to attribute additional values. For instance, in our example: "*vamos explodir o caveirão e capota o Águia*", the terms "explodir"("explode"—in English), "caveirão"("big skull"—in English), "*Águia*" ("eagle"—in English), "*capota*"("hood"—in English) have suspicious values associated individually to each term as specified in the ontology instances (e.g., "*Águia*"—1 and "big skull"—4), which are summed to constitute the post's suspicious level. When a new post is analyzed, an instance of *Act* class is generated; this instance is linked to other existing instances, which describe the CSE (terms). In this example, new acts (two facts of *Act(?ac01)* predicate) are generated and associated with other instances related to the terms (CSE) used in the post; then, rules 1 and 2 (cf. Listing 1) are executed. Consequently, the actions are linked to *Explode* class; they revive labels (i.e., *hasLabel* property with *blowUpPoliceCar* and *blowUpHelicopter* values) and suspicious levels (i.e., hasWeight property with "*8*" and "*6*" values). The values associated with these rules are then added to this post, increasing its suspicious level.

In the example, "*Trás cimento para eu dar um pico*" (presented in Section 4.1.1) the terms "*cimento*" ("cement" —in English) and "*dar um pico*" ("giving a peak" —in English) generate instances of an act related to *Cocaine* executing the rule 3 (Listing 1) and, consequently, linking it with *DrugUse* class. The Weight of 6 is given to this action, which increases the suspicious level of the post.

Some ontology design decisions were made when choosing the use of SWRL, as well as we defined a procedure to model and use the rules. The main reason we chose to use SWRL is due to its high-level abstract syntax for Horn-like rules. We consider this syntax adequate to represent how combinations of concepts, present in posts, can be evaluated (in the rule body/antecedent) to infer criminal acts suspicious (in the rule head/consequent). Complex rules can be defined by using SWRL, including chained inferences about new facts, preparing the framework for further improvements, even if the current version uses simpler and more direct rules.

In the framework, these rules can be fired using the following procedure: (1) new acts (instances of the Act class—Act(?ac01) are created when a post is analyzed by the framework (using Protégé API); (2) the framework reads the post and it links (using objectProperties) this act according to the terms used in the post (using Protégé API); (3) a reasoner (http://www.hermit-reasoner.com/) evaluates the rules and, if the new action matches with its body, it associates the act with criminal actions and respective weights; and, finally, (4) the framework reads the weights and attributes a suspicious level to the post. In the current version, the new instances (from the analyzed posts) can be saved for debugging and storage proposals.

The current set of rules was created from the evaluation of two researchers (co-authors) of the terms used in the ontology. Although it is impracticable to create rules for all possible combinations, this is an extendable model that can evolve into an increasingly representative set over time, through the analysis of new tweets and the concepts of the ontology.

## 5. A Case Study on Twitter®

This section presents the conducted case study based on posts from Twitter® to evaluate the feasibility of the framework. Section 5.1 focuses on the use of ML techniques for automatic intention classification of posts with CSE expressions. Section 5.2 presents a prototype implementing the OFCIC. We also describe scenarios used in the evaluation of the prototype for selecting suspicious messages and filtering them according to illocution classes.

### 5.1. OFCIC for the Detection of Criminal Expression and Intentions

We present our studies of how to use ML algorithms to automatically classify posts according to the illocution classification framework. The objective is to analyze to what extent some of the most used ML techniques can support the automatic classification of posts in the nature under analysis in this work. Section 5.1.1 presents the process used to construct the dataset (for training and test) and to evaluate the effectiveness of the ML techniques. It is very important to mention that, to the best of our

knowledge, there are no datasets ready to use in the context of this investigation (social media post with CSE). In this sense, it was necessary to manually create our own dataset. Section 5.1.2 presents the results of the execution of the ML techniques exploring the constructed datasets.

### 5.1.1. Procedures and Dataset

Our evaluation process started by the construction of the training/validation/test dataset, as presented in Figure 3. The social media messages used in this evaluation came from Twitter® posts. To create our dataset, we collected 8,835,016 tweets as our input (A1 in Figure 3). The dataset is composed by two parts: the first part is the same dataset used in the ontology validation (the last section), containing 5,896,275 tweets (274 tweets used before were excluded), which were collected in the first week of October 2018. The second part includes 2,938,741 tweets collected in the third week of August 2019. These tweets contain at least one term potentially related to CSE, collected from a large number of tweets (A1 in Figure 3).

The dataset was filtered according to steps A2 to A5 of Figure 3, resulting in 704 tweets, which were translated (according to our technique) and used as our training and testing dataset using a cross-validation method [49] for avoiding bias. Only highly suspicious tweets (i.e., 704 tweets) were used for training and testing our ML techniques because, in our framework (Figure 4), these techniques are used only to predict classes of intentions of posts based on CSE. In this sense, we judged input non-CSE related posts for training purposes as not necessary. The entire set of tweets (i.e., 8,835,016 tweets) was used to evaluate the steps A1, A2, and A3 of the OFCIC Training Component (Figure 3) and steps B1, B2, and B3 of the OFCIC Intention Classification Component (Figure 4). The 704 tweets were used to train and test the ML model (A4, A5, A6, A7, and A8 in Figure 3) and to evaluate the results (B4, B5, B6, B7, and B8 in Figure 4). Therefore, ML results refer to cross validation (and 80–20 split) using 704 tweets.

Once the tweets are collected, we selected the tweets with higher probability of being related to criminal expressions to adjust the ontology weights (A2 and A3 in Figure 3). To this end, a weight value (relevance level) was assigned to each term represented in the OntoCexp. This value was assigned in an iterative process by the involved researchers, as described in the last section. Seven iterations were performed between 9 September and 17 October 2019. Weights were assigned by one of the authors and revised by two others at each interaction.

We adopted the following procedure to assign suspicious values. During the first iteration, each ontology CSE term (instances dataproperties) had their values inversely adjusted to their occurrence in the initial dataset. For instance, *"açúcar"* ("sugar" in the English language) and "farinha" (*"flour"* in the English language) are very common terms used to refer to drugs, but they are much more frequent in normal tweets in the analyzed dataset. These terms are in the first fifth of most frequent words in tweets among the terms represented in the ontology, so we attributed the weight 1 to these terms, whereas the crime specific terms received higher values. For instance, *"Vai de vala"* ("go to the ditch" in the English language) is not frequent in normal (Portuguese language) tweets. This term received initially the weight 5, since they are among the 20% less frequent in the tweets.

The iterative process was made up of adjustments according to the researchers' interpretation of the selected sentences (i.e., using the previously assigned values). When a non-suspicious post was selected, the researchers analyzed the reasons and decreased the suspicious weight value of the terms linked to this post. For instance, the term *"Águia"* ("eagle" —in the English language) is not frequent in the initial dataset of tweets and received a high suspicious value (5); after the analysis, the researchers reduced the value to 1, since there were selected posts referring to eagle (animal). Weights were also assigned to the combinations of terms (modeled by the SWRL rules). For instance, when a post includes "eagle" and other terms (e.g.,"let's explode" and "to hood"), this is a highly suspicious post.

After setting the weight values to the terms, 39,120 tweets with highest values (using CSE weight and SWRL rules) were selected for manual analysis. In addition, 1044 tweets were then considered to

have a high potential to be related to criminal activity. Finally, duplicates were removed and the set of 704 was considered (A4 in Figure 3).

In the next step (A5 in Figure 3), the sentences (tweets) were manually classified by a researcher (first author) and cross-reviewed by two others. Our training/test dataset includes (cf. Table 1): 267 inducements, 150 assertions, 116 proposals, 78 valuations, 54 wish, 21 forecasts, and 16 contrition, totalizing 704 tweets. No message (in our case each tweet has only one message) was classified as palinode.

**Table 1.** Number of tweets by illocution classes.

| Class | # |
|---|---|
| Inducements | 267 |
| Assertions | 150 |
| Proposals | 116 |
| Valuations | 78 |
| Wish | 54 |
| Forecasts | 21 |
| Contrition | 16 |
| Palinode | 0 |
| **Total** | **704** |

Afterwards, two datasets were submitted for ML algorithms training and testing. The first one contains the 'Original Phrases' and the second one automatically 'Deciphered Phrases' (both datasets contains 704 Phrases). This process translates key-phrases on the original messages using the OntoCexp, each sentence is transformed in a set of tokens, and the translation is processed token-by-token or grouped as key-phrases. (the source code is available at https://github.com/ricardoresende/SlangWordsTranslation).

For the training phrase, we explored four word embedding techniques pre-trained for the Portuguese language in [86]. They were used to produce numerical representation of the posts, and then to use these representations in ML techniques. We can define word embedding as vectors of real numbers representing words in an *n*-dimensional space. The objective was to numerically represent the syntactic, semantic and morphological aspects of textual datasets. Several algorithms have been proposed to generate word embeddings, among which we highlight the following: Global Vectors (Glove) [87], Word2Vec/Word2Vec_Skip [88], Wang2Vec/WangVec_Skip [89], and FastText/FastText_Skip [90]. We adopted and evaluated in our study these four algorithms due to two reasons: (1) they are among the most recent used techniques to generate word embeddings; and (2) there is a repository available of the algorithms trained for the Portuguese language [86]. This contains vectors of 50, 100, 300, 600, and 1000 dimensions.

We chose a real number of vectors composed of 600 dimensions because this size presented the best cost-benefit in our case (a vector of 1000 dimensions did not improve the results, as demonstrated in [86] for similar tasks). Thus, to produce a feature vector, each word in a phrase is replaced by the numeric vector composed of 600 real values of a given word embedding. Typically, ML techniques work as fixed numbers of attributes for the entire dataset. In our context, phrases present a variable number of word totals. Thus, we defined the dimension of the feature vector according to the sentence with the largest number of words multiplied by 600 for each dataset. This resulted in 9000 dimensions for Original Phrases and 11,400 for Deciphered Phrases datasets. Shorter phrases have the remaining vector positions completed with 0. If a word is not available in pre-trained word embedding, this word is changed to the numeric vector with each letter *e*.

We used a scikit-learn package (Machine Learning in Python) (https://scikit-learn.org/stable/)—Accessed in January 2020. as the implementation of the SVM, Neural Network, Random Forest, and Naive Bayes classifiers [91]. This package allows for setup of several parameters to produce a classification model.

Initially, we considered four ML algorithms listed according to our review (cf. Section 3). The Naive Bayes presented the worst results in our preliminary assessments (0.41 of F1-Score for the best confirmation using a deciphered dataset). In addition, we experienced difficulties of implementing it with the techniques of word embedding. Therefore, our study focused on further investigating SVM, Neural Network, and Random Forest. Section 5.1.2 presents the obtained results. In the following, we detail the considered configurations for these algorithms:

- **ANN:**—We use the *MLPClassifier* (http://scikit-learn.org/stable/modules/generated/sklearn. neural_network.MLPClassifier.html—Accessed in January 2020) class as our solution for Artificial Neural Networks (ANN). This class implements an ANN of Multilayer Perceptron (MLP) type and allows training the neural network with Backpropagation algorithm. An ANN is characterized as MLP when it satisfies two criteria: the structure of the neural network presents at least one intermediate layer, also known as hidden layer, and uses a nonlinear activation function for the neurons. The Backpropagation algorithm performs a comparison between the achieved results and the expected results in the neural network output layer. From this last layer, the algorithm adjusts the synaptic weights of the network. Conceptual details about MLP and the backpropagation algorithm can be consulted in [49].

  The main parameters of the *MLPClassifier* class are: *alpha*, *hidden_layer_sizes* and the activation function. The *alpha* parameter is a scalar value. The *hidden_layer_sizes* parameter indicates the number of neurons in the intermediate layers, and it is necessary to specify the total number of neurons for each intermediate layer. The activation function selected was the hyperbolic tangent (Equation (1)):

  $$f(x_j) = tanh(x_j) = \frac{e^{x_j} - e^{-x_j}}{e^{x_j} + e^{-x_j}}, \qquad (1)$$

  where $x_j$ is the input value of the $j$ neuron, for $1 \leqslant j \leqslant N$, and $N$ is the total number of neurons in the neural network.

  We set $alpha = 0.1$ and trained the ANN for one and three intermediate layers. The following ANN configuration was used for the one intermediate layer case:

  – **Input Layer**: Total neurons equals to feature vector dimension;
  – **Intermediate layer**: composed of $L$ neurons;
  – **Output layer**: 8 neurons (one for each illocution class).

  In the case of three intermediate layers, two intermediate layers were added to the above-mentioned structure: the second intermediate layer, composed of $L/2$ neurons and the third one, also composed of $L/2$ neurons. The *hidden_layer_sizes* parameter was set to the respective value of each mentioned hidden layer. We trained and evaluated the following values for $L$: $20 \leqslant L \leqslant 300$, for every $L$ multiple by 20.

- **SVM:** We used the *SVC* (https://scikit-learn.org/stable/modules/generated/sklearn.svm.SVC. html—Accessed in January 2020) class as our solution for SVM. This class implements the training and classification of separable nonlinear data by combining a *kernel* function and a linear SVM. In particular, the *fit* and *predict* methods of the *SVC* class are responsible for training and classification, respectively. Thus, from the received parameters, the *SVC* class maps the dataset to an abstract space using a *kernel* function and then linearly separates the data looking for the maximum margin hyperplane. We use the multi-class version *one-versus-one*. Linear and nonlinear SVM theory, as well as kernel functions, which can be consulted in detail in [49].

  The key parameters of the *SVC* class were: the kernel function and C (error penalty parameter). We chose the kernel function Radial Basis Function (RBF), defined in Equation (2). It was also necessary to determine a value for the $\gamma$ parameter of the kernel function:

  $$K(\mathbf{x}_i, \mathbf{x}_j) = e^{-\gamma \left\| \mathbf{x}_i - \mathbf{x}_j \right\|^2}, \qquad (2)$$

where $\mathbf{x}_i$ e $\mathbf{x}_j$ are two vectors of the dataset and $\gamma \geq 0$ is a parameter whose value should be adjusted during training.

We evaluated during training the following parameter settings: $10^{-3} \leqslant C \leqslant 10^2$, for every $C$ multiple by 10. For each value of $C$, $10^{-3} \leqslant \gamma \leqslant 10^2$, for every $\gamma$ multiple by 10.

- **Random Forest**—We used the *RandomForestClassifier* (https://scikit-learn.org/stable/modules/generated/sklearn.ensemble.RandomForestClassifier.html—Accessed in January 2020) class as our solution for Random Forest. The main parameter is the *n_estimators* (the number of trees in the forest). We adopted *n_estimators* $= 100$ for all tests performed, after a preliminary analysis of values between 20 and 160, for every multiple of 10.

We adopted the default values for the parameters not mentioned according to the respective implementations. We used the Synthetic Minority Over-sampling Technique (SMOTE) as class balancing technique to generate synthetic samples for all classes, but the majority class [92]. During training, all classes present the same number of samples. Importantly, the SMOTE technique was applied only to the training dataset and not to the test dataset. In the case of applying the cross-validation technique, for each new data division, SMOTE was applied only for training folds, not being applied to the test fold. We used the SMOTE implementation provided by [93].

### 5.1.2. Results of Intention Detection Using ML Algorithms

Tables 2–4 present the experimental results obtained with the ANN, SVM, and Random Forest classifiers. In these analyses, we used the cross-validation method with $k = 5$, i.e., divided the data set into 5 folds. We performed 12 times cross-validation to choose the best parameter values. The column **Best Configuration** in these tables indicates the values of the main parameters of the implementation of each classifier that resulted in the most generalist model possible among the values of the parameters tested. For each of the algorithms assessed, the parameters presented are:

- **ANN**: number of intermediate layers (1 or 3) / value of $L$;
- **SVM**: $g$ / $c$;
- **Random Forest**: *n_estimators*.

Table 2 presents overall results (considering average results of all illocution classes). The average F1-Scores presented in Table 2 were calculated in two steps: (1) for each cross-validation step, the class-weighted overall F1-score is calculated; (2) at the end of the five rounds, we calculated the average F1-score of these five F1-scores. In Tables 3 and 4, we present the average F1-Score per class, i.e., the F1-Score of each class is the average of the five F1-scores calculated for each respective class during cross-validation.

For the deciphered dataset, we adopted the following criteria for choosing values as the best parameters in order of priority of choice: (1) correctly classifying at least one sentence into seven classes of intent and overall average F1-Score greater than 0.30; or (2) overall average F1-Score greater than 0.30 and whose result of the classifier had the lowest number of illocution classes with an overall average F1-Score equal to zero. For the original dataset, we produced the results using the best parameters defined for the deciphered dataset. This is justified because a preliminary analysis indicated that, regardless of the parameter values within the range presented above, the overall average F1-Score for the original dataset was always lower than the experimental results obtained with the deciphered dataset. In addition, for ANN, sometimes two or more parameter values resulted in the same average F1-Score. In these cases, we adopted as the first criterion of choice the smallest number of intermediate layers (1 instead of 3), and, as the second tiebreaker rule, the smallest value of $L$, i.e., the smallest number of neurons.

The experimental results in Table 2 indicate that the best results obtained were an overall average F1-Score equal to 0.45 (ANN), 0.47 (SVM), and 0.45 (Random Forest) using the deciphered phrases. For the original phrases, the experimental results obtained were an overall average F1-Score equal

to 0.40 (ANN), 0.38 (SVM), and 0.43 (Random Forest). We found that the results are improved when we use the automatically decrypted phrases compared to the original phrases extracted from Twitter. For the original phrases, the best performance was obtained with the Random Forest classifier, whereas, for the deciphered phrases, the three classifiers presented practically a similar performance of the overall average F1-Score. In the case of SVM, we observe higher variations according to the Word Embedding technique, which is not observed in the results presented by the ANN and Random Forest classifiers.

By analyzing the experimental results in Tables 2 and 3, we observe that, although the best overall average F1-Score was presented by the SVM classifier for the deciphered data 0.47 (cf. Table 2), results in Table 3 indicate that the ANN classifier was the only one that did not result in a F1-Score equal to zero for the Contrition illocution class considering all Word Embedding techniques. Results highlighted in italic indicate the classifier did not assign any sentences to that illocution class. Results highlighted in bold indicate the best values obtained for a given illocution class, by a classification technique, considering all Word Embeddings techniques.

Table 2 shows that the overall average F1-Score with the ANN and Random Forest classifiers did not present large differences according to the word embedding technique, unlike the SVM classifier. In the case of the SVM classifier, we obtained an overall average F1-Score equal to 0.47 with Word2Vec_skip, whereas, with FastText, it was only 0.30. For the ANN and Random Forest classifiers, the best result was 0.45, whereas the worst was 0.44. This shows that the SVM classifier is further affected by the word embedding techniques in our dataset.

The average F1-Score by class in Tables 3 and 4 presented significant differences depending on the word embedding technique. As an example, the case of the *Forecast* class presented an average F1-Score equal to 0.25 with Word2Vec_skip and only 0.06 with Glove (cf. Table 3) for the ANN classifier. When we used the original phrases (cf. Table 4), the same behavior was observed. For example, the case of *Forecast* class for the SVM classifier: average F1-Score equals to 0.27 with Word2Vec and 0 with FastText (Table 4). In addition, we observed that the best average F1-Score (equals to 0.63) was obtained for the Inducement class, using the SVM classifier, the Word2Vec_skip technique, and deciphered phrases. We found that the used word embedding techniques influence the results of classification.

As mentioned earlier in this section, we performed 12 times the cross-validation technique with $k = 5$. For each execution of the cross-validation method, a new random distribution of phrases was performed to produce the five folds. Defining $k = 5$, approximately 80% of the phrases is used for training, corresponding to four folds, and approximately 20% is used for the test. The exact value depends on the test fold, since with 704 phrases available, one of the folds has two more phrases than the others.

Table 5 presents the best results obtained from all 60 training/test executions performed using the dataset divided into two sets of 80% and 20% for training and testing, including the results by class and the overall value for F1-Score and Accuracy. By analyzing Table 5, the best overall F1-Scores obtained were: 0.51 and 0.54 (w/o contrition class- Table 5) for ANN, with seven and six non-zero F1-Score classes, respectively; 0.55 for SVM, but with six classes with non-zero F1-Score; 0.49 and 0.52 with Random Forest, with seven and six non-zero F1-Score classes, respectively. Considering F1-Score by class, the best result was 0.72 for the Inducement class, with the SVM classifier and the Word2Vec technique.

**Table 2.** Overall experimental results obtained with the ANN, SVM, and Random Forest classifiers. Word embedding vector dimension: 600. Cross-validation method with $k = 5$.

| ML Technique | Word Embedding Technique | Best Configuration | Overall Average F1-Score Original Phases | Overall Average F1-Score Deciphered Phase |
|---|---|---|---|---|
| ANN | fastText | 1/240 | 0.39 | 0.44 |
| | fastText_skip | 1/160 | 0.38 | 0.45 |
| | Word2Vec | 1/220 | 0.39 | 0.44 |
| | Word2Vec_skip | 1/280 | 0.39 | 0.45 |
| | Wang2Vec | 1/200 | 0.38 | 0.44 |
| | Wang2Vec_skip | 1/140 | 0.38 | 0.45 |
| | GloVe | 1/120 | 0.40 | 0.44 |
| SVM | fastText | 0.01/1 | 0.27 | 0.30 |
| | fastText_skip | 0.01/1 | 0.37 | 0.44 |
| | Word2Vec | 0.01/1 | 0.37 | 0.45 |
| | Word2Vec_skip | 0.01/1 | 0.38 | 0.47 |
| | Wang2Vec | 0.01/1 | 0.27 | 0.31 |
| | Wang2Vec_skip | 0.01/1 | 0.37 | 0.43 |
| | GloVe | 0.01/1 | 0.35 | 0.39 |
| Random Forest | fastText | 100 | 0.43 | 0.45 |
| | fastText_skip | 100 | 0.43 | 0.44 |
| | Word2Vec | 100 | 0.41 | 0.44 |
| | Word2Vec_skip | 100 | 0.41 | 0.44 |
| | Wang2Vec | 100 | 0.41 | 0.45 |
| | Wang2Vec_skip | 100 | 0.41 | 0.45 |
| | GloVe | 100 | 0.41 | 0.44 |

**Table 3.** Experimental results obtained with the ANN, SVM, and Random Forest classifiers with Deciphered Phases. Word embedding vector dimension: 600. Cross-validation method with $k = 5$. Average F1-score by illocution class .

| ML Technique | Word Embedding | Best Configuration | Overall Average F1-Score by Illocution Class (Deciphered Phases) | | | | | | | |
|---|---|---|---|---|---|---|---|---|---|---|
| | | | Proposal | Inducement | Forecast | Wish | Assertion | Valuation | Palinode [1] | Contrition |
| ANN | fastText | 1/240 | 0.36 | **0.60** | 0.08 | 0.40 | 0.44 | 0.22 | - | 0.04 |
| | fastText_skip | 1/160 | **0.38** | 0.58 | 0.08 | 0.43 | 0.40 | **0.35** | - | **0.15** |
| | Word2Vec | 1/220 | 0.36 | **0.60** | 0.16 | 0.36 | 0.40 | 0.27 | - | 0.07 |
| | Word2Vec_skip | 1/280 | **0.38** | 0.59 | **0.25** | **0.47** | 0.4 | 0.29 | - | 0.07 |
| | Wang2Vec | 1/200 | 0.33 | 0.58 | 0.23 | 0.43 | 0.41 | 0.30 | - | 0.07 |
| | Wang2Vec_skip | 1/140 | 0.37 | 0.59 | 0.11 | 0.44 | **0.45** | 0.24 | - | 0.10 |
| | GloVe | 1/120 | 0.37 | 0.57 | 0.06 | 0.48 | 0.41 | 0.34 | - | 0.08 |
| SVM | fastText | 0.01/1 | 0.19 | 0.56 | *0* | 0.12 | 0.19 | 0.05 | - | *0* |
| | fastText_skip | 0.01/1 | **0.39** | 0.60 | 0.07 | **0.38** | 0.39 | 0.24 | - | *0* |
| | Word2Vec | 0.01/1 | 0.38 | 0.61 | 0.17 | 0.31 | 0.42 | **0.29** | - | *0* |
| | Word2Vec_skip | 0.01/1 | **0.39** | **0.63** | **0.26** | 0.34 | **0.45** | 0.28 | - | *0* |
| | Wang2Vec | 0.01/1 | 0.19 | 0.56 | *0* | 0.16 | 0.18 | 0.08 | - | *0* |
| | Wang2Vec_skip | 0.01/1 | 0.37 | 0.61 | 0.07 | 0.37 | 0.39 | 0.24 | - | *0* |
| | GloVe | 0.01/1 | 0.30 | 0.58 | *0* | 0.31 | 0.31 | 0.23 | - | *0* |
| Random Forest | fastText | 100 | 0.36 | 0.61 | 0.21 | 0.41 | 0.41 | 0.30 | - | *0* |
| | fastText_skip | 100 | 0.33 | 0.59 | 0.15 | 0.44 | **0.43** | 0.25 | - | 0.06 |
| | Word2Vec | 100 | 0.33 | 0.59 | 0.30 | **0.51** | 0.39 | 0.22 | - | *0* |
| | Word2Vec_skip | 100 | **0.37** | 0.59 | 0.28 | 0.40 | 0.38 | 0.25 | - | **0.08** |
| | Wang2Vec | 100 | 0.34 | **0.62** | 0.21 | 0.39 | 0.39 | 0.29 | - | *0* |
| | Wang2Vec_skip | 100 | 0.29 | 0.61 | **0.36** | 0.38 | 0.42 | **0.31** | - | 0.05 |
| | GloVe | 100 | 0.37 | 0.59 | 0.31 | 0.42 | 0.42 | 0.21 | - | 0.05 |

[1] In the samples, there are no tweets previously classified in the Palinode intent class.

**Table 4.** Experimental results obtained with the ANN, SVM, and Random Forest classifiers with the Original Phases. Word embedding vector dimension: 600. Cross-validation method with $k = 5$. Average F1-score by illocution class.

| ML Technique | Word Embedding | Best Configuration | Average F1-Score by Illocution Class (Original Phases) | | | | | | | |
| --- | --- | --- | --- | --- | --- | --- | --- | --- | --- | --- |
| | | | Proposal | Inducement | Forecast | Wish | Assertion | Valuation | Palinode [1] | Contrition |
| ANN | fastText | 1/240 | 0.29 | 0.52 | 0.03 | 0.30 | **0.41** | 0.23 | - | 0.06 |
| | fastText_skip | 1/160 | 0.27 | 0.51 | 0.14 | **0.36** | 0.40 | 0.23 | - | **0.15** |
| | Word2Vec | 1/220 | 0.32 | 0.51 | 0.24 | 0.32 | 0.38 | **0.28** | - | *0* |
| | Word2Vec_skip | 1/280 | 0.33 | 0.51 | **0.25** | 0.35 | 0.33 | 0.27 | - | 0.2 |
| | Wang2Vec | 1/200 | 0.29 | **0.53** | 0.18 | 0.25 | 0.36 | 0.26 | - | *0* |
| | Wang2Vec_skip | 1/140 | 0.30 | 0.52 | 0.03 | 0.32 | 0.39 | 0.21 | - | 0.09 |
| | GloVe | 1/120 | **0.38** | **0.53** | 0.18 | 0.31 | 0.39 | 0.21 | - | 0.10 |
| SVM | fastText | 0.01/1 | 0.06 | **0.55** | *0* | 0.16 | 0.09 | 0.16 | - | *0* |
| | fastText_skip | 0.01/1 | 0.27 | 0.51 | 0.09 | 0.35 | 0.39 | 0.16 | - | *0* |
| | Word2Vec | 0.01/1 | 0.28 | 0.47 | **0.27** | 0.32 | 0.39 | **0.28** | - | *0* |
| | Word2Vec_skip | 0.01/1 | **0.29** | 0.50 | 0.18 | 0.37 | **0.40** | 0.22 | - | *0* |
| | Wang2Vec | 0.01/1 | 0.03 | 0.54 | *0* | 0.14 | 0.15 | 0.17 | - | *0* |
| | Wang2Vec_skip | 0.01/1 | 0.26 | 0.53 | 0.08 | **0.39** | 0.33 | 0.20 | - | *0* |
| | Glove | 0.01/1 | 0.16 | 0.52 | 0.08 | 0.33 | 0.34 | 0.25 | - | *0* |
| Random Forest | fastText | 100 | 0.36 | 0.55 | 0.33 | 0.38 | **0.43** | 0.24 | - | **0.10** |
| | fastText_skip | 100 | **0.37** | **0.57** | 0.28 | 0.39 | 0.37 | **0.27** | - | **0.10** |
| | Word2Vec | 100 | 0.25 | 0.56 | 0.32 | **0.46** | 0.40 | 0.24 | - | **0.10** |
| | Word2Vec_skip | 100 | 0.26 | **0.57** | **0.35** | 0.33 | 0.42 | 0.16 | - | **0.10** |
| | Wang2Vec | 100 | 0.30 | 0.54 | 0.31 | 0.44 | 0.38 | 0.19 | - | 0.08 |
| | Wang2Vec_skip | 100 | 0.25 | **0.57** | 0.36 | 0.40 | 0.37 | 0.23 | - | 0.08 |
| | GloVe | 100 | 0.30 | 0.56 | 0.29 | 0.43 | 0.39 | 0.23 | - | 0.10 |

[1] In the samples there are no tweets previously classified in the Palinode intent class.

**Table 5.** Experimental results obtained with the ANN, SVM, and Random Forest classifiers with Deciphered Phases. Word embedding vector dimension: 600. Dataset divided into two sets of 80% and 20% for training and testing, respectively. The Weighted average F1-score column was calculated as a function of the number of tweets by each class.

| ML Technique | Word Embedding | Best Configuration | F1-Score by Illocution Class (Deciphered Phases) | | | | | | | | | |
| --- | --- | --- | --- | --- | --- | --- | --- | --- | --- | --- | --- | --- |
| | | | Proposal | Inducement | Forecast | Wish | Assertion | Valuation | Palinode [1] | Contrition | F1-Score | Accuracy |
| ANN | fastText | 1/200 | **0.50** | 0.56 | 0.36 | 0.36 | 0.48 | **0.42** | - | 0.40 | 049 | 0.49 |
| | fastText_skip | 1/160 | 0.39 | 0.59 | 0.33 | 0.55 | **0.54** | 0.41 | - | 0.29 | **0.51** | **0.51** |
| | Word2Vec | 1/220 | 0.41 | **0.71** | 0.22 | 0.26 | 0.39 | 0.23 | - | 0.33 | 0.49 | 0.48 |
| | Word2Vec_skip | 3/120 | 0.45 | 0.65 | 0.25 | 0.38 | 0.45 | **0.38** | - | 0.33 | 0.50 | 0.51 |
| | Wang2Vec | 1/280 | 0.36 | 0.60 | **0.46** | 0.52 | **0.49** | 0.32 | - | 0.33 | 0.50 | 0.49 |
| | Wang2Vec_skip | 1/140 | 0.47 | 0.61 | 0.20 | **0.70** | 0.45 | 0.15 | - | **0.50** | 0.49 | 0.50 |
| | GloVe | 1/120 | 0.42 | **0.61** | 0.20 | 0.43 | 0.45 | **0.21** | - | **0.40** | **0.47** | 0.46 |
| ANN(w/o all classes) | Wang2Vec | 1/120 | **0.49** | **0.66** | **0.33** | 0.46 | 0.59 | 0.37 | - | *0* | **0.54** | **0.54** |
| SVM | fastText | 0.01/1 | 0.30 | 0.57 | *0* | 0.43 | 0.18 | 0.12 | - | *0* | 0.35 | 0.45 |
| | fastText_skip | 0.01/1 | 0.48 | 0.69 | **0.25** | **0.58** | 0.37 | 0.36 | - | *0* | 0.51 | 0.54 |
| | Word2Vec | 0.01/1 | **0.55** | **0.72** | **0.29** | 0.11 | 0.43 | 0.30 | - | *0* | 0.50 | 0.53 |
| | Word2Vec_skip | 0.01/1 | 0.43 | 0.65 | **0.29** | 0.56 | **0.62** | 0.41 | - | 0 | **0.55** | **0.56** |
| | Wang2Vec | 0.01/1 | 0.40 | 0.58 | *0* | 0.13 | 0.18 | 0.11 | - | 0 | 0.35 | 0.44 |
| | Wang2Vec_skip | 0.01/1 | 0.41 | 0.56 | **0.25** | 0.47 | 0.52 | 0.18 | - | *0* | 0.46 | 0.47 |
| | Glove | 0.01/1 | 0.43 | 0.62 | 0 | 0.42 | 0.46 | 0.32 | - | 0 | 0.47 | 0.51 |
| Random Forest | fastText | 100 | 0.48 | 0.67 | 0.14 | 0.32 | **0.56** | 0.29 | - | *0* | **0.52** | **0.51** |
| | fastText_skip | 100 | 0.38 | 0.64 | 0.29 | 0.42 | 0.49 | 0.24 | - | **0.29** | 0.49 | 0.50 |
| | Word2Vec | 100 | 0.23 | **0.69** | 0.33 | **0.55** | 0.55 | 0.38 | - | *0* | **0.52** | **0.52** |
| | Word2Vec_skip | 100 | 0.31 | 0.61 | **0.67** | 0.42 | 0.48 | **0.52** | - | 0 | 0.50 | 0.49 |
| | Wang2Vec | 100 | 0.47 | 0.62 | 0.29 | 0.32 | 0.54 | 0.24 | - | *0* | 0.48 | 0.49 |
| | Wang2Vec_skip | 100 | **0.56** | 0.65 | 0.25 | 0.42 | 0.44 | 0.21 | - | *0* | 0.49 | 0.50 |
| | GloVe | 100 | 0.48 | 0.64 | 0.25 | 0.26 | 0.49 | 0.31 | - | *0* | 0.49 | 0.49 |

[1] In the samples, there are no tweets previously classified in the Palinode intent class.

## 5.2. The OFCIC Framework in Use

We present the implementation and usage scenarios of the SocCrime web application. This prototype implements the OFCIC framework for automated selection of suspicious posts and filtering according to intention classes. Our software implements the elements described in Figure 4 by including elements adjusted according to the studied Twitter® dataset, such as: the ontology with weight values resulted from the selection of posts; and trained models used to classify input posts according to the intention classes.

Figure 7 presents the search interface of SocCrime prototype. In this interface, users can select keywords and intention classes. The prototype presents two main components: the first is responsible for selecting and indexing the posts; and the second is responsible for searching the pre-selected posts.

The first component receives/accesses a set of posts to be analyzed (e.g., using the Twitter® search API available at [79]). The solution uses the ontology OntoCexp (with weight values already defined) to automatically select the suspicious posts. In the sequence, these posts are automatically classified using the ML algorithms (with an already trained model). Finally, these messages are included in an indexing search service or a database. It is out of the scope of the prototype to implement a new search service. The SocCrime prototype was implemented to connect to an existing service using JSON or a database interface.

The second component performs searchers by accessing a search service using the keywords selected by the users at the search interface (Figure 7). The SocCrime prototype receives the results by filtering them according to the selected intention. Finally, as Figure 8 shows, the results are presented in a report interface. These results contain a rank from the most suspicious post to the less suspicious ones (the bottom of Figure 8). The order is determined according to the weights of the CSE and rules of OntoCexp and the intention class was filtered according to ML results.

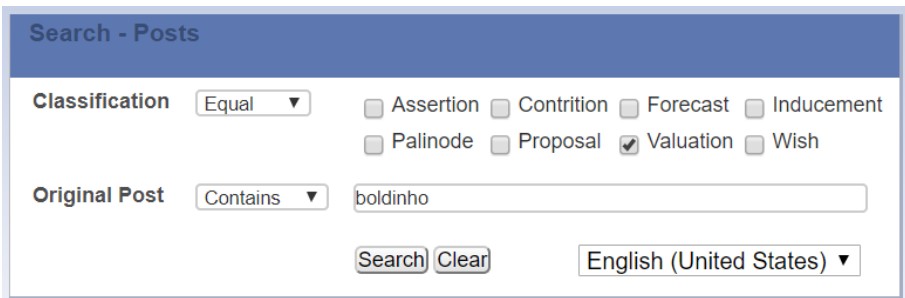

**Figure 7.** Search interface of the SocCrime Prototype.

| | Classification | Code | Twitter Id | Original Post |
|---|---|---|---|---|
| 🔍 | Valuation | 204,369 | 1161427960810942466 | Caralho que boldinho filé.... |
| 🔍 | Valuation | 204,355 | 1161779703092527106 | Era só o boldinho nesse frio 😞 |
| 🔍 | Valuation | 204,389 | 1160975829138137098 | Era só um boldinho 😞 |
| 🔍 | Valuation | 204,383 | 1161063782669541376 | Era só um boldinho 😞 |
| 🔍 | Valuation | 204,306 | 1162898968369647619 | " Era só um boldinho 🌹 msm.. |
| 🔍 | Valuation | 3,147,329 | 1047372402865426432 | Esse boldinho deu um sono bacana rs |
| 🔍 | Valuation | 204,377 | 1161319376068448258 | @gomeserpidio E hj tu é ,'contra' a maconha e posta que quer boldinho kkkkkkkk lokaaaáaaaa a senhora |

[1 to 7 of 11]　　Go to 1　View 7

**Figure 8.** Search result report interface of the SocCrime Prototype.

In the following paragraphs, we present one result from each of the seven illocution classes from the Liu's framework [22,23] identified (based on the automatic classification) with our solution. This illustrates the different interpretation of the recovered posts according to each illocution class.

The *Contrition* class allows for identifying, for instance, when someone intends to say regrets or apologizes for something in the past. The post *"Minha boca tá rasgada de tanto loló. Nunca mais"*

(My mouth is cut because the abuse of loló (an aerosol of a chloroform-based drug). I will never use loló again—in the English language—translated by the authors) represents that the user regrets the abuse of this drug. This class allows, for instance, to distinguish messages that express regret from messages that propose the use of drugs. These messages can be interesting, for example, in campaigns to sensitize users about advantages to stop the use of drugs.

The post *"meteram bala nos verme, explodiram o caveirão, detonaram a cabine, mataram 5 alemão"* (bulletted the worm, blew up the big skull, detonated the cabin, killed 5 Germans—in the English language—translated by the authors) is an assertion about past facts describing that: a police officer has killed (bulletted the worm), an armored police vehicle was blown up (blew up the big skull), a police booth was destroyed (detonated the cabin), and five people from the *Favela do Alemão* were killed (five Germans killed). In this case, the facts were described, without the attribution of values, or planning of something.

The post *"Amiga cuidado com as drogas ok . . . marijuana td bem mas cocaina ja eh demais"* (. . . my friend beware of drugs ok . . . marijuana is ok but cocaine is too much—in the English language—translated by the authors) refers to a valuation. In this post, someone compares two drugs and recriminates the consume of strong drugs, possibly addressed to someone who used these drugs.

The post *"café com marijuana ou o café com acido?"* (coffee with marijuana or coffee with acid?—in the English language—translated by the authors) is a proposal. In this post, someone makes a question about which drug the user will consume the next morning.

The post *"Chico doce vai te abraçar em !!!"* (Chico doce will hug you !!!—in the English language—translated by the authors) is an inducement. In this post, someone makes a threat (or warning) to another person about the possibility of aggression (hug you) using a police baton (*Chico doce*).

The post *"se der um teco vai virar o caos"* (if I give a beat will turn chaos—in the English language—translated by the authors) is a forecast. In this post, someone analyzes what will happen if (s)he consume illegal drugs (give a beat).

The post *"quero pó e loló de cafe da manha"* (I want to have powder and loló in the breakfast—in the English language—translated by the authors) is a wish. In this post, someone expresses the desire of consuming cocaine (powder) and loló.

These examples illustrate the framework's ability to retrieve and classify relevant posts of all studied ilocution classes. The main limitations of the framework is related to retrieving and classifying relevant posts with indirect speech acts [94], and those are context-dependent. For instance, the post *"você deve levar mais açucar"* (you must take more sugar—in the English language—translated by the authors) apparently is not a suspicious post; however, information about the context (e.g., who spoke and situation) can turn it into a suspicious one, since the term açucar ("sugar") can be related to cocaine.

## 6. Discussion

Expression of intentions is a key element in human communication. This applies to criminal communication using social media. Criminals, like the rest of the population, take advantage of Web facilities and online social networks in their activities [1]. Computational solutions considering data analysis can be used as a tool for criminal investigation and prevention. Currently, the huge quantity of message, speed, and complexity of the digital environment turns the manual search and analysis of posts from social networks with criminal content impracticable.

This work goes one step further towards the improvement of computational support in tasks considering criminal intentions of posts using criminal slang (deliberately or unintentionally ciphered messages). Our literature review demonstrated that existing investigations mostly emphasize sentiment analysis tasks. Indeed, there is a lack of software tools for analyzing, representing, and detecting criminal intention in social media posts (cf. Section 3). Our literature review provided us with strategies on how to deal with the problem based on the careful analysis of related tasks.

Existing studies (e.g., [60,63,74]) showed that a strategy that combines techniques (e.g., ontologies and ML) is a more viable path to deal with our problem.

We proposed a framework (OFCIC) and defined an ontology (OntoCexp) to support the formal representation of criminal vocabulary. Our solution explored the OntoCexp ontology and ML techniques. This provided a viable way to build the SocCrime software prototype that automatically selects posts written using crime slang and translates them to "standard language". The framework is suited to automatically classify posts' intentions. In a simplified interface, users can choose keywords and illocution classes; then, the prototype returns a report with suspicious posts. Scenarios illustrated the differences in results when considering each class of illocution.

The OFCIC framework is complex and involves various steps and technologies. The extensive use of OFCIC may demand computational and human resources for its refined implementation, including key steps such as: (1) collection of social media posts for the construction of the training model; (2) adjustments to the weights of ontology terms and the definition of additional and improved rules to be used in the post selection; and (3) ML training (according to the technique used). However, it is important to note that this does not impact the scalability of the framework in use. Once the selection of the posts is performed using OntoCexp with the previously adjusted weights, the classification occurs based on the pre-trained model. The posts are included in a conventional search engine (which performs keyword-based retrieval), and the posts are filtered according to pre-classified list in linear time.

One key aspect to be considered is the long-term maintenance and evolution of the OntoCexp. The current version of the ontology presents 165 Classes, 635 individuals, 3410 Axioms, 1533 Logical Axioms, 848 Declaration Axioms, and 44 Object Properties, resulting in ALCHO(D) DL expressivity [95]. Crime domain is dynamic and complex. CSE is constantly evolving and rationalized. CSEs described in Mota [12] are a good starting point, and presented positive results by selecting relevant posts in our Twitter® study. Nevertheless, the OntoCexp might be expanded (to represent other regional CSE) with adequate updating techniques. This includes the adjustments on the weights of the terms used by OFCIC. The ontology updating efforts are focused on the inclusion of new instances (i.e., new terms) and SWRL rules. Further studies on how to support (or automate) these tasks are needed. This might include, for instance, techniques for automatic ontology evolution.

The conducted case study with Twitter® shows, in practice: (1) the viability of the framework execution; (2) the expressiveness and utility of the ontology; and (3) the analysis of effectiveness in applying the ML techniques in our context. Although a relatively large number (8,835,290) of tweets was considered for the case study, this study must be expanded to include a larger time window. This may result in more tweets used for ML training and adjusting the weight of terms in the ontology. This may reach better results and an expanded study of this subject. In addition, aspects such as CSE regionalism and studies in other countries/languages can be executed.

This study can be expanded to other online social networks. Other social media, for example, with more restricted communication channel can be used by criminals in other ways. Studies with these networks can contribute to the evolution and refinement of our framework and ontology. It is important to mention that Twitter® maintains a public access API, which facilitated the study reported in this article.

Another aspect to be discussed is the effectiveness of the investigated ML techniques. The best ML configurations presented F1-score around 0.5 (in the cross validation). This number is close to those obtained in [96] for special education domain, which uses elaborated sentences written in standard language (consider a more formal written style without slang).

We highlight some aspects in the analysis of the obtained results concerning the ML models. First, the results refer to a multiclass classification with eight classes, which makes the classification problem harder as compared with typical binary classification problems. The individual results of the classes show that, in specific situations, in which the user is interested in a specific class, this number is higher, for example, 0.71 was obtained for the inducement class. In these cases, we can improve the

results by using multiple binary classification, i.e., a class is individually trained against the others. We obtained 0.72 of general F1-Score by combining GloVe and ANN techniques. In this case, the user cannot select multiple classes at once.

Results can be further improved by increasing our training set. In addition, 704 tweets may be too little for the cross validation, especially when we consider that some classes have very few (e.g., contrition) or no samples (e.g., palinode). Despite the balancing techniques explored in our investigation (SMOTE [92]), larger and balanced training sets may enhance the achieved results. Long-term studies should provide larger and further representative training sets.

In addition, other approaches, such as the use of CNN and RNN, already trained in the transfer learning technique can be explored to improve the ML results. For example, we can investigate how to improve the results by combining CNN and GRU techniques [50]. One approach would be to use trained CNN/RNN (with millions of messages) for sentiment analysis, and use in conjunction with other classifiers. Other approaches, such as transforming vectors (result of word embedding) into images and using ML solutions for this purpose, can be explored. We emphasize that the trained networks available for binary classification are more abundant than 8 classes classification, and their suitability to the problem of intention classification is the subject of future research.

We found positive results with the use of deciphered phrases by the applied ontology as compared with original phases. In general, deciphered phrases increased the score by 0.02–0.09 in most of the configurations assessed. Considering experimental results obtained with cross-validation (Table 2), we obtained 0.05 of increase on average from 0.38 to 0.43 (around 13% of improvement) (The Student's *t*-test (two dependent paired samples, with distribution close to the normal curve) including all configuration values of Table 2 resulted on $t = 10.67$ and $p$-value $< 0.0001$ (two-tailed), showing significant improvement considering 0.01 significance level (i.e., 99% of confidence)). When we consider the best configuration (may be the most interesting in practice), we obtained 0.09, from 0.38 to 0.47 using SVM Word2Vec_skip 0.01/1 (around 24% of improvement), which is not negligible in terms of increasing the measures of ML techniques. A plausible explanation to this fact is that the word embedding algorithms were trained for Portuguese messages (in the standard form). The main fact that justifies this translation is its potential of reusing techniques developed for standard writing language, enabling new possibilities in long-term research.

Our research adopted a word-by-word translation strategy, which explored the structure of the OntoCexp. This ontology can be combined with advanced translation techniques, which combines ontologies with statistical and ML techniques. This can result in improved translation and the increase of the ML scores. This strategy requires additional research focused on translation of slang and ciphered text to the standard ones. Therefore, the obtained results can be considered a baseline for an OFCIC framework implementation, which provides room for improvements in further research. The efforts to maintain the ontology as well as alternatives to minimize these efforts must be investigated when comparing future translation techniques.

The SocCrime prototype showed the viability of the OFCIC execution and use. Additional evaluations of the use of the framework in practical settings are still required. In fact, how it influences the practice of human investigators and their impacts should come from a long-term study, which requires a system in production and agreements with law enforcement agencies. From a practical point of view, one of the challenges is to increase the obtained results from the ML techniques. The overall result around 0.5 is relatively low, and even individual class results (it may be what matters in practice and are higher, as discussed) may not be the expected for law enforcement agencies. However, the illocution classification is an additional tool to provide filtering and ranking options for suspicious messages selected by the framework. Despite the current results, these tools can be useful in practice when we consider the huge number of social network posts, which makes manual assessment practically impossible. In addition, investigators may be frequently interested only in highly suspicious posts, which are in the first positions of the rank and are more likely to be recovered by the framework, due to the occurrence of CSE.

Finally, we indicate that this is a pioneering and promising study on how ontologies and ML techniques improve the support for the analysis of intentions of social media posts, making use of CSE. The results obtained from the conducted case study point out the viability of the framework, with immediate contributions in the selection of suspicious messages from social network posts. This article opened further research directions for improving each part of our solutions (framework, ontology, ML, and prototype).

## 7. Conclusions

Social media have become an instrument for planning and executing crimes. In this sense, new investigation and prevention software tools must be studied and proposed. Intention expression must be considered in these tools for supporting investigators to select and analyze suspicious social network posts because intention refers to an important element of human communication. In this article, we proposed the framework OFCIC for supporting expert users in this task. We employed a multiple-strategy approach to deal with the automatic selection and classification of written posts from online social networks. Our developed framework explored the OntoCexp as a formal and extensible ontology applied to help identifying posts with CSE on social networks and decipher these posts based on slang vocabulary. We found effective the use of ML techniques in OFCIC for automatic classifying posts according to illocution classes. Our research contributes with an innovative framework to support the investigation of suspicious criminal posts on social media. The presented case study with real tweets demonstrated the viability of the proposal and challenges to be addressed in future research. Our future work includes evolving the OntoCexp to be more comprehensive by including additional concepts and CSEs (e.g., regionalism and other languages). We plan to investigate techniques to (semi)automatically update our ontology based on social media data. In addition, we aim to assess other ML techniques (e.g., deep learning) and build a larger ML training dataset. This work opens perspectives to refine our developed SocCrime software prototype to be employed and evaluated in practical settings, including the use of advanced search mechanism and visualization techniques.

**Author Contributions:** Conceptualization, R.R.d.M., D.F.d.B., F.d.F.R., J.C.d.R. and R.B.; methodology, R.R.d.M., F.d.F.R., J.C.d.R., and R.B.; software, R.R.d.M. and D.F.d.B.; validation, R.R.d.M. and D.F.d.B.; formal analysis, F.d.F.R.; investigation, R.R.d.M. and D.F.d.B.; resources, F.d.F.R.; data curation, R.R.d.M.; writing—original draft preparation, R.R.d.M., D.F.d.B., F.d.F.R., and R.B.; writing—review and editing, F.d.F.R., J.C.d.R., and R.B.; supervision, F.d.F.R., J.C.d.R., and R.B.; funding acquisition, J.C.d.R. and R.B. All authors have read and agreed to the published version of the manuscript.

**Funding:** This research was funded by Coordenação de Aperfeiçoamento de Pessoal de Nível Superior (CAPES) Grant No. 88882.366274/2019-01, Ministry of Science, Technology, Innovation and Communications PCI Program Grant No. 444303/2018-9 and São Paulo Research Foundation (FAPESP) (Grant #2017/02325-5) (The opinions expressed in this work do not necessarily reflect those of the funding agencies).

**Conflicts of Interest:** The authors declare no conflict of interest.

## Abbreviations

The following abbreviations are used in this manuscript:

| | |
|---|---|
| ANN | Artificial Neural Networks |
| CNN | Convolutional Neural Networks |
| CSE | Criminal Slang Expression |
| DTDNN | Distributed Time Delay Neural Network |
| GRN | Gated Recurrent Networks |
| ML | Machine Learning |
| MLP | Multilayer Perceptron |
| NLP | Natural Language Processing |
| OFCIC | Ontology-Based Framework for Criminal Intention Classification |
| OntoCexp | Ontology of Criminal Expressions |
| OWL | Ontology Web Language |
| TF-IDF | Term Frequency-Inverse Document Frequency |
| LSTM | Long Short-Term Memory |
| RBF | Radial Basis Function |
| RDF | Resource Description Framework |
| RNN | Recurrent Neural Network |
| SAT | Speech Act Theory |
| SMOTE | Synthetic Minority Over-sampling Technique |
| SRL | Semantic Role Labeling |
| SVM | Support Vector Machine |

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
