# Peer review of "A Framework for Detecting Intentions of Criminal Acts in Social Media: A Case Study on Twitter†"

_information, doi:10.3390/info11030154_

Round 1

Reviewer 1 Report

The paper presents an approach to analyse and classify intentions in Tweets, potentially using criminal slang language. The proposed method first uses an ontology to translate the messages into "standard language" and then uses a trained ML model to classify them into one of 8 intention classes. While the task is interesting and an important one, and the paper is reasonably well written, unfortunately there are several important aspects missing, or not treated in enough detail in the paper. Hence, the paper would need considerable improvement to be published. Since my expertise is in the area of ontologies, and ontology engineering, my review tends to focus on these parts, while I hope that other reviewers have gone into detail on the use of ML.

My first question to the authors is: why are you using an ontology? Towards the end of the paper the authors conclude that translating the posts to standard language increases the F-score by 0.02-0.09, which is not a huge contribution. Is that a statistically significant increase in all cases? Further, that just means that the translation contributes, but why do it through an ontology then? What about other methods. The rest of the parameters of the setup are tested quite throughly, so why not this one? Why not several methods of translation, e.g. some by the ontology and some ML ones? It would be nice to see how they compare, and if it is really worth the effort of maintaining a large terminology/ontology, or if it would be easier/better to just retrain a ML model continuously.

Further, regarding the ontology: It is not clear to me what the ontology is actually used for. The ontology that I can find on the github link references is nothing near what is described in the paper. It does not contain all the classes that are shown in the examples, it has very few axioms included (and several of them seem incorrect, e.g. the use of several ranges for object properties, which implies intersection of the classes and not union), and it is very sparsly commented. Is this an early version of the ontology? Then please link to the correct version from the paper, so that this can be studied along with the paper. Why not publishing the ontology online, i.e. according to semantic web practices, instead of just on github, if it is anyway in RDF/OWL? The paper also needs a better explanation of the task of the ontology. How are weights represented in it? How are they retrieved, updated, used? How are the rules represented in relation to the ontology? I guess another rule language is used for these? Or SPARQL constructs? More specifically, when studying Figure 8, it is not clear to the reader what was there in the ontology to start with and what things gets populated when a sentence is analysed, and rules are executed. This should be shown step by step. It is also not clear why new instances need to be produced, i.e. what is the role of the blowUpPoliceHelicopter (or just blowUpHelicopter?) instance that is produced? Why is that needed? And how are all these instances related back to the text of the message to produce the translation?

Additional questions arise around the datasets used. The large numbers of collected Tweets are mentioned several times, which is a bit misleading, since in the end the dataset used is only around 700 Tweets. In particular, the paper needs to explain and characterise this dataset much more closely: does this only contain Tweets that have manually been confirmed to be about criminal activity? In that case it would be a very biased dataset. A more realistic dataset should be used, where most of the Tweets are NOT about criminal activity, reflecting the actual distribution on Twitter. Or else the skewed dataset must be very carefully motivated. Additionally, how are the intention categories distributed over the dataset, i.e. how many Tweets of each category? Are all Tweets in one category or is there/should there be an "other things" category, for things that do not map to these 8 intention classes? The characteristics of the dataset should be given in absolute numbers, and also I miss some absolute numbers (in addition to F-scores) in the evaluation results, to give a more intuitive feeling about the results. In addition to numbers it is also important to analyse the results further, and not only in terms of what the system gets right, but in particular what it gets wrong. A careful analysis of the mistakes would give much insight into the limitations of the approach, and giving some typical negative examples to the reader would be very useful.

Regarding the results, of course I can imagine that intention classification is a hard task, but are these numbers really enough to claim it is useful? It does not seem that any user evaluation has been made, and in fact the use case presented does not seem to be targeted at end users, e.g. police, since such a user interface seems highly inappropriate for Law Enforcement Agencies (LEA). LEAs are interested in investigating crimes, and not seeing statistics or lists of Tweets of a certain category. I would imagine that LEAs might be interested in Tweets at a certain time point, from a certain geographic location, or from within a restricted set of socially connected Twitter users, and then to from there analyse their intentions at the moment - not some general listings of Tweets with certain intentions. Overall, the paper would be greatly improved if the authors could show some evidence towards the usefulness of the resulting framework.

Another issue in the paper, although more on the theoretical side of the paper, is the survey part. First of all, what is the purpose of this survey? In the paper it is presented as a structured literature review in order to retrieve related work. However, considering the search keywords, I think that the survey does not really find all related work. It merely finds work that has employed exactly the same kind of framework, i.e. ontology-based work, which deals with intention/semiotics etc., and specifically in the crime domain. Actually, I am a bit surprised that so many ML frameworks were retrieved based on that search query, since now ML-related keywords were used. But in particular I find it problematic to restrict the domain of application, i.e. crime domain, since this work could easily apply to another domain, hence much related work could probably be found elsewhere. As related work I would rather classify any approach attempting to solve the same, or a similar problem, i.e. to analyse and classify Tweets containing slang language into multiple categories that are rich representations of meaning, i.e. not simply binary classification. I understand that this is a much harder search to do, but also a much more valuable one for the paper. Hence, I would rather suggest to just skip the formal survey method and try to find any such methods "by hand", but then to compare to these properly, i.e. in terms of accuracy and effectiveness, etc. Rather than just superficially describing a set of work that have potentially inspired this one. As it is now the related work section is quite unclear - how do all these approaches relate and compare to this one?

Additional detailed comments and questions, in order of appearance:
- There is an imbalance in section 2.4, where some approaches are described in much detail, while others are mentioned in brief.
- Lines 273-276: why only 100 from Google scholar? While you had several hundred from the other sources?
- Line 290: why exclude books?
- Section 3.1 and 3.2 need to explain how each approach relates to the one proposed here.
- Line 304: "sorted manually" - what does that mean?
- Line 330: significant improvement compared to what?
- Line 331: how was this paper retrieved by your query since it is about depression and not crime at all?
- Lines 400-401: how do you do training automatically?
- Lines 406-408: I don't understand what this describes. Extraction of an ontology by TFIDF?? Or what is extracted? And for what is it used?
- Line 416: Reduced performance compared to what? And what is "performance" in this case? The whole sentence is unclear.
- Line 425: NLP is usually done through ML these days, so what do you mean here?
- Line 443-446: I do not see what semiotics adds really? Do you use this practically somehow? How? What? This is not clear.
- Line 447-450: This is even more unclear, what is "pragmatics" and how does it relate to the paper, in practice?
- Line 468-469: What is function and what is content? Unclear! And how can function be classified automatically in the training phase, isn't this what you are training the model to do later on, by now using labelled data?
- Line 469-471: this whole process is completely opaque. What does the annotator do exactly? What are the weights? How are they represented? Stored? On what basis are they updated? Why are they in the ontology? See also general comments above.
- Line 495: "The adjustment..." - same as the last comment.
- Figure 5: why is the manual classification/labelling done before deciphering? What about mistakes introduced in deciphering? Change in meaning? How is that evaluated/taken into account?
- Line 499: Expressing rules for any possible combination of slang terms seems completely impossible, so what does this mean actually?
- Line 506: What selected posts? Selected how? And what does cross-check is recommended mean?
- Line 528-529: Here it sounds as the survey was about determining the frequency of use of certain algorithms, but this was not really what was presented in the survey section.
- Figure 6: steps B7 and B8 are not very clear. How does B7 use the ontology? There seem to be people involved, how? What is the outcome? A database, or also a user interface?
- Section 4.2.1: How was this process iterative? In what way? Was there no methodology used? How were ontology requirements specified? Where the ontology tested against the requirements?
- Line 581-583, 588-591: What about false positives? Did you get Tweets that were not actually about criminal activity? How many? What did you do with these? And the false negatives? It is not clear how either of these were used. Additionally, if you mean that you tuned the ontology to exactly these Tweets and then you used the same Tweets later in the evaluation, that is a severe flaw in the evaluation setup!
- Line 593-594: How should it be updated? What effort would it take? By whom? Is it realistic to even do that?
- Section 4.2.2: It is not clear how the descriptions relate to the categories, e.g. how do the things described under the heading "human beings" relate to human beings, i.e. the concept person? For instance, how are places related to people? Aggressions? Sexuality? How are criminal activities connected to criminal acts, and further to money? It is unclear what you mean by these "groups of concepts", and what are actually the concepts and relations in the ontology. Especially since the ontology as describe here cannot actually be found at reference [76].
- Figure 8: see general comments on the ontology, and lack fo description on how it is used.
- Listing 1: what do you mean by inference rules? Axioms that invoke OWL inferences? Rules expressed in another separate rule language (like SPARQL, SWRL)? Also please explain the rule syntax.
- Line 658: What doe "have weights associated individually" mean?
- Line 703-704: How can a tokenization process translate phrases??
- It is not clear to me how Table 4 relates to Tables 1-3.
- Line 871-874: Is this really what LEAs would want (see earlier comment)? And why is keyword search only against the original post? I would assume you would use the whole ontology in this case, to index the posts, i.e. both with translated words and slang?
- Page 25: The first paragraph in the list presents what class it discusses, but none of the subsequent ones do, so it is not clear what they exemplify at all. Here would be a good place to also look at negative examples, issues, limitations etc.
- Line 921: not clear what multi-strategy refers to.
- Line 936: I am not convinced that they perform well.
- Line 940: interesting to know, but why did you not present this when you talked about the ontology before? What are the axioms? What are they used for? And how is the terminology expressed if the ontology only has classes, properties and a few axioms? Is it through annotations? Instances?
- Line 973: what balancing techniques? I did not see any.
- Reference list: there are some incomplete and some incorrect references, please review and revise the list. For instance, Lecture Notes in Computer Science is not a journal name, it is the name of a book series, where each book has an individual title, and it is this title that should appear in the reference, not only LNCS. Reference 85 has the wrong name of the proceedings volume, it is a workshop not a conference etc.

Language issues:
Line 40: hug -> huge
Line 42: reformulate "turn difficult"
Line 421: reveled -> revealed?
Line 518: Coaine -> Cocaine
Line 666: in which extend -> to what extent
Line 864: set posts -> set of posts
Line 883: massages -> messages
Line 999: In this sence, does not fit, rephrase
Line 1001: select -> to select

Author Response

Dear reviewer, 

The reviewers’ comments on the first version of our manuscript were very helpful, and we believe our article improved thanks to them. 

Please, find below the response and changes for each comment. 

 The paper presents an approach to analyse and classify intentions in Tweets, potentially using criminal slang language. The proposed method first uses an ontology to translate the messages into "standard language" and then uses a trained ML model to classify them into one of 8 intention classes. While the task is interesting and an important one, and the paper is reasonably well written, unfortunately there are several important aspects missing, or not treated in enough detail in the paper. Hence, the paper would need considerable improvement to be published. Since my expertise is in the area of ontologies, and ontology engineering, my review tends to focus on these parts, while I hope that other reviewers have gone into detail on the use of ML.

My first question to the authors is: why are you using an ontology? 

Response: We clarified this aspect in the new version of the introduction: “... We chose to use an  ontology to describe aspects related to the language used to commit criminal acts. The use of ontologies  allows us to describe relationships and make inferences to determine weights related to suspicious messages, which cannot be represented by simple vocabularies, or other less structured knowledge  representation systems. In addition, ontologies are expandable and interoperable models on the Web, which can be (re)used by otherWeb systems.” 

Towards the end of the paper the authors conclude that translating the posts to standard language increases the F-score by 0.02-0.09, which is not a huge contribution. Is that a statistically significant increase in all cases? 

Response: We included the following paragraph in the discussion section:“...Considering experimental results obtained with cross-validation (Table 2), we obtained 0.05 of increase on average from 0.38 to 0.43 (around 13% of improvement)15. When we consider the best configuration (may be the most interesting in practice), we obtained 0.09, from 0.38 to 0.47 using SVM Word2Vec_skip 0.01/1 (around 24% of improvement), which is not negligible in terms of increasing the measures of ML techniques. A plausible explanation to this fact is that the word embedding algorithms were trained for Portuguese messages (in the standard form). The main fact that justifies this translation is its potential of reusing techniques developed for standard writing language, enabling new possibilities in long-term research.”  

The following footnote about the significance test of the increase was also included: “The Student’s t-test (2 dependent paired samples, with distribution close to the normal curve) including all configuration values of Table 2 resulted on t=10.67 and p-value<0.0001 (two-tailed), showing significant improvement considering 0.01 significance level (i.e., 99% of confidence).”

Further, that just means that the translation contributes, but why do it through an ontology then? What about other methods. The rest of the parameters of the setup are tested quite throughly, so why not this one? Why not several methods of translation, e.g. some by the ontology and some ML ones? 

Response: In fact, our research looked for alternatives for improving the translation, particularly using ontology and ML (combine both seemed most promising in this case). However, we did not find a ready-to-use solution in the context of slang expressions in Portuguese language. Therefore, we chose to use a more direct solution (using the ontology structure) to show the potentials of the proposed framework (focus of this paper), which also includes the application of the ontology in other stages. We might focus on the improvement of the translation algorithm in future research. We appreciate the comment, and modified our discussion section to include: “Our research adopted a word-by-word translation strategy, which explored the structure of the OntoCexp. This ontology can be combined with advanced translation techniques, which combines ontologies with statistical and ML techniques. This can result in improved translation and the increase of the ML scores. This strategy requires additional research focused on translation of slang and ciphered text to the standard ones. Therefore, the obtained results can be considered a baseline for a OFCIC framework implementation, which provides room for improvements in further researches. The efforts to maintain the ontology as well as alternatives to minimize these efforts must be investigated when comparing future translation techniques.”.    

It would be nice to see how they compare, and if it is really worth the effort of maintaining a large terminology/ontology, or if it would be easier/better to just retrain a ML model continuously.

Response: We agree with this comment because the effort to maintain the ontology should be further considered. We included a new statement in the discussion section about this concern. This comparison demands long term studies (probably years), such that we can precisely determine the efforts required in practice. We must also consider that the ontology is used in other features in the framework, including, for instance, the selection of posts for composing ML training sets. In addition, the formal representation of slang expressions can be valuable for other applications and human interpretation (which was not explored in this paper). 

Further, regarding the ontology: It is not clear to me what the ontology is actually used for. 

Response:  We included at the beginning of the subsection “4.2. OntoCexp - Ontology of Criminal Expressions” a summary of the functionalities that make use of the ontology:  “In summary, the OFCIC uses the OntoCexp in both components as follows: (1) Training Component (Figure 3) uses the OntoCexp to select (according to the use and combinations of CSE) suspicious posts to train and test the ML algorithms. The weights of CSE and acts are manually adjusted, as mentioned before. This ontology is further used to translate the posts; and, (2) the Intention Classification Component (Figure 4) uses the ontology to select suspicious posts (this is an automatic process, once the weights have already been adjusted) and to translate the messages by then submitting them for classification. This section presents a synthesis of the OntoCexp. ”             

The ontology that I can find on the github link references is nothing near what is described in the paper. It does not contain all the classes that are shown in the examples, it has very few axioms included (and several of them seem incorrect, e.g. the use of several ranges for object properties, which implies intersection of the classes and not union), and it is very sparsly commented. Is this an early version of the ontology? Then please link to the correct version from the paper, so that this can be studied along with the paper. Why not publishing the ontology online, i.e. according to semantic web practices, instead of just on github, if it is anyway in RDF/OWL?

Response: In fact, we made the mistake of sharing the wrong version (Commit 10/15/2018 11:09). The newest version of the ontology is now shared in GitHub. An updated version is also avalive in the Web Protégé Repository. 

The paper also needs a better explanation of the task of the ontology. How are weights represented in it? How are they retrieved, updated, used? How are the rules represented in relation to the ontology? I guess another rule language is used for these? Or SPARQL constructs? 

Response: We included more details about how the ontology. We used the SWRL to specify our rules, Listing 1 is now on standard SWRL syntax (instead a simplified syntax).  

More specifically, when studying Figure 8, it is not clear to the reader what was there in the ontology to start with and what things gets populated when a sentence is analysed, and rules are executed. This should be shown step by step. It is also not clear why new instances need to be produced, i.e. what is the role of the blowUpPoliceHelicopter (or just blowUpHelicopter?) instance that is produced? Why is that needed? And how are all these instances related back to the text of the message to produce the translation?

Response: We included several new explanations across the section 4.2 aiming to make clear for the reader, for instance: 

  • “As the objective of OntoCexp is to represent CSE, and not the entire crime domain, we made some design decisions focused on our goals. In our ontology, classes are concepts represented by CSE, while instances are terms (CSE) used to refer to these concepts. Each instance (CSE) has labels associated with variations in their writing form. Weights values are linked to these terms and refer to the degree of suspicion. All terms of crime vocabulary proposed in [13] were included in the ontology.” (end of subsection 4.2.1.)
  • “During the post selection steps, each CSE receives a value according to (iteratively) predefined weights. In addition, rules in the ontology are executed to attribute additional values. For instance, in our example: “vamos explodir o caveirão e capota o Águia”, the terms “explodir”(“explode” – in English), “caveirão”(“big skull” – in English), “Águia” (“eagle” – in English), “capota”(“hood” – in English) have suspicious values associated individually to each term as specified in the ontology instances (e.g., “Águia” - 1 and “big skull” - 4), which are summed to constitute the post’ suspicious level. When a new post is analyzed an instance of Act class is generated, this instance is linked to other existing instances, which describe the CSE (terms). In this example, new acts (two facts of Act(?ac01) predicate) are generated and associated with other instances related to the terms (CSE) used in the post, then rules 1 and 2 (cf. Listing 4.2.3) are executed. As consequence the actions are linked to Explode class, they revive a labels (blowUpPoliceCar and blowUpHelicopter) and suspicious levels (weights - "8" and "6"). The values associated with these rules are then added to this post, increasing its suspicious level.” (end of subsection 4.2.3.)

Additional questions arise around the datasets used. The large numbers of collected Tweets are mentioned several times, which is a bit misleading, since in the end the dataset used is only around 700 Tweets. In particular, the paper needs to explain and characterise this dataset much more closely: does this only contain Tweets that have manually been confirmed to be about criminal activity? In that case it would be a very biased dataset. A more realistic dataset should be used, where most of the Tweets are NOT about criminal activity, reflecting the actual distribution on Twitter. Or else the skewed dataset must be very carefully motivated. 

Response:  In the new version, we included the following paragraphs, which better explain our dataset: “To create our dataset, we collected 8,835,016 tweets as our input (item A1 of Figure 3). The dataset is  composed by two parts: the first part is the same dataset used in the ontology validation (last section), containing 5, 896, 275 tweets (274 tweets used before were excluded), which were collected in the first  week of October 2018. The second part includes 2, 938, 741 tweets collected in the third week of August 2019. These tweets contain at least one term potentially related to CSE, collected from a large number of tweets (A1 in Figure 3).

The dataset was filtered according to the steps A2 to A5 of Figure 3, resulting in 702 tweets, which were translated (according to our technique) and used as our training and testing dataset using  cross-validation method [48] for avoiding bias. Only highly suspicious tweets (i.e., 702 tweets) were used for training and testing our ML techniques because in our framework (Figure 4) these techniques are used only to predict classes of intentions of posts based on CSE. In this sense, we judged not necessary input non CSE related posts for training purposes. The entire set of tweets (i.e., 8, 835, 016 tweets) was used to evaluate the other framework steps.”. 

Additionally, how are the intention categories distributed over the dataset, i.e. how many Tweets of each category? Are all Tweets in one category or is there/should there be an "other things" category, for things that do not map to these 8 intention classes? The characteristics of the dataset should be given in absolute numbers, and also I miss some absolute numbers (in addition to F-scores) in the evaluation results, to give a more intuitive feeling about the results.

Response: According to Liu's framework (our theoretical basis used), all the messages (speech acts) are classified in one of the classes. We rephrased the presentation of the class distribution of our training/test dataset and included a new table: “Our training/test dataset includes (cf. Table 1):  267 inducements, 150 assertions, 116 proposals, 78 valuations, 54 wish, 21 forecasts, and 16 contrition, totalizing 704 tweets. No message (in our case each tweet has only one message) was classified as palinode.”   

In addition to numbers it is also important to analyse the results further, and not only in terms of what the system gets right, but in particular what it gets wrong. A careful analysis of the mistakes would give much insight into the limitations of the approach, and giving some typical negative examples to the reader would be very useful.

Response: We included an example and the main limitation of the framework at the end of Section 5: “These examples illustrate the framework’s ability to retrieve and classify relevant posts of all studied ilocution classes. The main limitations of the framework is related to retrieve and classify relevant posts with indirect speech acts [93], and those are context-dependent. For instance, the post “você deve levar mais açucar” (you must take more sugar – in English language – translated by the authors) apparently it is not a suspicious post, however information about the context (e.g., who spoke and situation) can turn it a suspicious one, since the term açucar (“sugar”) can be related to cocaine.” 

Regarding the results, of course I can imagine that intention classification is a hard task, but are these numbers really enough to claim it is useful? It does not seem that any user evaluation has been made, and in fact the use case presented does not seem to be targeted at end users, e.g. police, since such a user interface seems highly inappropriate for Law Enforcement Agencies (LEA). LEAs are interested in investigating crimes, and not seeing statistics or lists of Tweets of a certain category. I would imagine that LEAs might be interested in Tweets at a certain time point, from a certain geographic location, or from within a restricted set of socially connected Twitter users, and then to from there analyse their intentions at the moment - not some general listings of Tweets with certain intentions. Overall, the paper would be greatly improved if the authors could show some evidence towards the usefulness of the resulting framework.

Response: In Section 5, we aim to order Tweets, identifying and highlighting those most suspected of being related to crimes, or even terrorist events. The following phrases were included before the figures of Subsection 5.2: "These results contain a rank from the most suspicious post to the less suspicious ones (bottom of the Figure 8). The order is determined according to the weights of the CSE and rules of OntoCexp and the intention class was filtered according to ML results.". 

We agree with the comment on the need for end-user evaluations and field tests. However, a full and reliable user evaluation can be the subject of a new article (focused on this aspect) in the future. We share with you (below), our previous experiences with Brasilizan LEA, which inspired this work. We also further highlighted the utility of our solution in the Introduction of the paper.    

Brazilian LEAs used a critical system to support intelligence activities, in order to anticipate terrorist events (e.g., shootings, explosions, poisonings, etc.) in places with a high agglomeration of tourists. Our mission was to develop a technological solution to support intelligence activities aiming to integrate the activities of collecting and analyzing data from open sources on the Internet - especially social media (e.g., Tweeter, Facebook, Pastebin, etc.). As the collected data were anonymized and retrieved with some delay, we were able to determine some approximate geographic locations, in addition to restricted sets of suspicious users. These suspicions data and meta-data were sent to a task-force compounded of police officers and intelligence agents. A user interface, adapted to the LEAs needs, was developed in conjunction with the task-force. Information resulting from the system was classified as “secret” and this made academic publications impossible. This work was supported in part by the Information Systems Security Division at Renato Archer Information Technology Center (CTI) under Grant #PRJ4.35 H1 14/01088.  Preliminary results of the approach were used during big events held in Brazil (FIFA World Cup 2014 and Summer Olympics Games 2016).

Another issue in the paper, although more on the theoretical side of the paper, is the survey part. First of all, what is the purpose of this survey? In the paper it is presented as a structured literature review in order to retrieve related work. However, considering the search keywords, I think that the survey does not really find all related work. It merely finds work that has employed exactly the same kind of framework, i.e. ontology-based work, which deals with intention/semiotics etc., and specifically in the crime domain. Actually, I am a bit surprised that so many ML frameworks were retrieved based on that search query, since now ML-related keywords were used. But in particular I find it problematic to restrict the domain of application, i.e. crime domain, since this work could easily apply to another domain, hence much related work could probably be found elsewhere. As related work I would rather classify any approach attempting to solve the same, or a similar problem, i.e. to analyse and classify Tweets containing slang language into multiple categories that are rich representations of meaning, i.e. not simply binary classification. I understand that this is a much harder search to do, but also a much more valuable one for the paper. Hence, I would rather suggest to just skip the formal survey method and try to find any such methods "by hand", but then to compare to these properly, i.e. in terms of accuracy and effectiveness, etc. Rather than just superficially describing a set of work that have potentially inspired this one. As it is now the related work section is quite unclear - how do all these approaches relate and compare to this one?

Response:  Our proposal with a focused systematic literature review was to further understand the achievements in literature concerning intention detection in the crime domain. The specific aspect regarding the use of slang turns such context challenging. Our goal was to clarify the existing approaches to such problematic. We agree that additional studies concerning other domains can be valuable to improve this review (out of the scope of the paper). In order to improve the literature review, we added some manually selected papers (section 2.2) and further discussed how existing approaches are related to our approach. The presentation of the papers in section 3 were improved, as well as we highlighted  our contributions at the end of section 3.  

Additional detailed comments and questions, in order of appearance:

- There is an imbalance in section 2.4, where some approaches are described in much detail, while others are mentioned in brief.

Response: As also suggested by the other reviewer, we removed part of the explanation and figures related to ANN (which were not essential) to make this section more focused and balanced.

- Lines 273-276: why only 100 from Google scholar? While you had several hundred from the other sources? 

Response: We included the following footnote: “The google scholar returned thousands of results in a relevance  order, however no relevant titles were found between the 100th and 200th results. Consequently, we decided to stop the evaluation in the 100th paper.”

- Line 290: why exclude books?

Response: This part of the review protocol was included to avoid reviews of publications of different nature, e.g., research journal papers versus textbooks or collections of papers (in this case the papers were individually analysed). We included this footnote in the new version: “Paper collection books had their articles evaluated individually.”

- Section 3.1 and 3.2 need to explain how each approach relates to the one proposed here.

Response: We provided further discussions regarding to the originality of our proposal in section 3.3

- Line 304: "sorted manually" - what does that mean?

Response: The sentence was rephrased as follows : “... each slang expression is manually annotated with meanings assigned to it.”

- Line 330: significant improvement compared to what?

Response: The sentence was rephrased as follows: “According to the authors, the accuracy of the ML sentiment analysis classifiers were improved by using expert domain-specific knowledge from social networks.”

- Line 331: how was this paper retrieved by your query since it is about depression and not crime at all?

Response: This paper was retrieved from springerlink database using the query presented at our paper (probably because criminals are explicitly cited in the paper). It meets the inclusion criteria I1. In the final analysis, we considered of interest due to the used techniques and the link that the authors made with criminal or offending activities.     

- Lines 400-401: how do you do training automatically?

Response: The sentence was rephrased as follows: “Due to the lack of data in supervised learning, the authors proposed to automatically annotate tweets with labels inferred from crime trends. These annotations were the basis to automatically create training examples to be used by prediction models.”

- Lines 406-408: I don't understand what this describes. Extraction of an ontology by TFIDF?? Or what is extracted? And for what is it used?

Response: The sentence was rephrased as follows: “They extracted bag-of-words feature sets represented as TF-IDF (Term Frequency - Inverse Document Frequency). These sets were used to compare bot and human tweets. Then, they applied sentiment analysis using the LabMT tool to better understand the nature of words used by bots and humans.”

- Line 416: Reduced performance compared to what? And what is "performance" in this case? The whole sentence is unclear.

Response: The sentence was rephrased as follows: “In general, the studies presented better assertiveness measures in the categorization of sentiment using complementary approaches (e.g., semantic descriptors with ML) as compared with a single one. Although this represents a decrease of response time, the benefits of adopting complementary techniques may outweigh the computational cost.”

- Line 425: NLP is usually done through ML these days, so what do you mean here?

Response: The sentence was rephrased as follows: “the authors point out the benefits of adopting linguistics features in combination with ML techniques.”.

- Line 443-446: I do not see what semiotics adds really? Do you use this practically somehow? How? What? This is not clear.

Response: We cited Semiotics because this is the basis of the illocutionary act theory explored in our approach. Indeed, this is only indirectly explored in this work. However, for a matter of theoretical consistency we decided to leave the short description related to Semiotics in the paper. 

- Line 447-450: This is even more unclear, what is "pragmatics" and how does it relate to the paper, in practice?

Response: Semiotics together with the SAT are the theoretical basis for the illocution classification framework used in this work. We looked for other articles that had a similar approach in the CSE context (as related works), and they were not found. Pragmatics can be understood as a subfield of semiotics (and linguistics), and focuses on understanding the relationship of context and meaning, including intentions. It is the foundation used by many linguists in the evaluation of intentions in texts (manually). We included additional explanations in section 2.3 and section 3.3. 

- Line 468-469: What is function and what is content? Unclear! And how can function be classified automatically in the training phase, isn't this what you are training the model to do later on, by now using labelled data?

Response:  Indeed, the training set was manually generated.  We provided a new explanation: “The Liu’s framework is used to classify messages’ function part, specifying the illocution that characterizes the  speaker’s intent (i.e., we classify the function as one of the six illocution classes). The OntoCexp is used to classify the terms used in the content part, which represents the meaning of the message. We first  generated a training and test set by manually assigning post’ functions to one of the illocution classes. Once trained, ML techniques are used to automatically assign an illocution class to posts.”.

- Line 469-471: this whole process is completely opaque. What does the annotator do exactly? What are the weights? How are they represented? Stored? On what basis are they updated? Why are they in the ontology? See also general comments above.

Response:  We provided a new explanation: “During the training phase, terms of the content part are associated with concepts of the OntoCexp. Each term (individuals) belongs to a class (which defines a concept). Each term has a dataproperty weight value, which defines the chance of a term being related to CSEs. These weights are manually and interactively defined by the training component (cf. Figure 2 (A)), and they are used to select suspicious posts (cf. Figure 2 (B)).”

- Line 495: "The adjustment..." - same as the last comment.

Response:  We provided a new explanation: “In this component, the weights (i.e., chance of a term being related to CSE) are iteratively adjusted. A researcher (or domain specialist) defines weights, uses them to select posts, and adjusts them according to the relevance of the selected posts.”

- Figure 5: why is the manual classification/labelling done before deciphering? What about mistakes introduced in deciphering? Change in meaning? How is that evaluated/taken into account?

Response: In our case, the researchers (some of coauthors) that worked in the manual classification are able to read and interpret the CSE, thus it is preferred to work with original phases. This classification can also be done after deciphering, but this will be affected by mistakes introduced by the deciphering mechanisms. We included new content at the discussion about the advances and drawbacks of the deciphering mechanism.     

- Line 499: Expressing rules for any possible combination of slang terms seems completely impossible, so what does this mean actually?

Response: As it is really impossible to produce rules with all combinations, the quality of the selection falls on the modelers' ability to write the most relevant rules to define suspicious phrases. The iterative process of modeling and successive adjustments as well as the evolutionary characteristics of web ontologies minimize (but do not eliminate) this problem. The need for improvement of rules definitions (for extensive use of OFCIC) is now mentioned at the discussion section.  

- Line 506: What selected posts? Selected how? And what does cross-check is recommended mean?

Response: The sentences were rephrased as follows: “In order to build a labeled training base to be employed in ML algorithms (Figure 3 (A7)) and create the training model (Figure 3(A8)), at this step we manually assigned illocution types (Figure 3(A5)) for the posts included in our training set ((A4) in Figure 3). It is desirable that the assigned illocution types be checked by other researchers (or domain specialists).”

- Line 528-529: Here it sounds as the survey was about determining the frequency of use of certain algorithms, but this was not really what was presented in the survey section.

Response: The sentence was rephrased as follows: Our systematic review (Section 3), which is focused on ontologies, intentions and crime, shows several investigations using ML, of which the most frequent ones are those mentioned above.”

- Figure 6: steps B7 and B8 are not very clear. How does B7 use the ontology? There seem to be people involved, how? What is the outcome? A database, or also a user interface?

Response: We included new explanations about step B7 (inclusion and use of search service), as well as a reference of a paper detailing how it can be accomplished: “According to the adopted search service, ontologies can be used to improve the results. Authors in [78] details how the search mechanisms can be improved by using intention and query expansion techniques based on ontologies.”

We also included new explanations about step B8: “Finally, an query and analysis interface is provided (Figure 4 (B8)). This may include search interfaces, reports and visualization techniques. Section 5 presents a search interface prototype whose results (posts) are presented in a rank according to the suspicious level and intention class.”

 - Section 4.2.1: How was this process iterative? In what way? Was there no methodology used? How were ontology requirements specified? Where the ontology tested against the requirements?

Response: We included several sentences in this section related to this comment. This included additional explanations about methodology, requirements, scope and evaluation. 

- Line 581-583, 588-591: What about false positives? Did you get Tweets that were not actually about criminal activity? How many? What did you do with these? And the false negatives? It is not clear how either of these were used. Additionally, if you mean that you tuned the ontology to exactly these Tweets and then you used the same Tweets later in the evaluation, that is a severe flaw in the evaluation setup!

Response: This evaluation is a step in the iterative construction of ontology (section 4.2.1). In this step we are not interested in statistics about the false negative (or positive), since ontology was under initial construction. Ontology is a generic model to represent CSE that had pre-existing vocabularies as its primary source (e.g., Mota 2016). The objective was to provide feedback for the modelers about how the terms are used in practice (including the analysis of false positive and negative). The ontology was not tuned to this set of 274 posts, and this set was not used in the case study described in section 5. We included additional explanations in section 4.2.1.

- Line 593-594: How should it be updated? What effort would it take? By whom? Is it realistic to even do that?

Response: The task of maintaining vocabularies with updated CSE is performed by institutions, regardless of the existence of our ontology. Thus, the ontology updating efforts are focused on the inclusion of new instances (i.e., new terms) and SWRL rules. Our design decision to use SWRL to express the combination of CSEs took into account the ease (in our view) of keeping the rules up to date as compared with internal classes axions. The discussion about the ontology maintenance was improved (section 6).  

- Section 4.2.2: It is not clear how the descriptions relate to the categories, e.g. how do the things described under the heading "human beings" relate to human beings, i.e. the concept person? For instance, how are places related to people? Aggressions? Sexuality? How are criminal activities connected to criminal acts, and further to money? It is unclear what you mean by these "groups of concepts", and what are actually the concepts and relations in the ontology. Especially since the ontology as describe here cannot actually be found at reference [76]. Figure 8: see general comments on the ontology, and lack for description on how it is used.

In Section 4.2.2, the ontology (OntoCexp) was updated to represent the relations of some classes aiming to improve the expressivity. We reviewed the names of categories, e.g., "Human beings" to "Person". To improve the readability, we structured the textual description of the concepts and we call "groups", i.e., we grouped terms considered similar or that can be related and described in the same group and paragraph in the paper. The updated version of OntoCexp (V3) is available in a long-established source of ontologies (Protégé Web) (in addition to GitHub), as suggested. After the modifications in the paper, the reference number was updated from 76 to 84. We included a brief description on how the ontology is used: "The ontology requirements were defined by the researchers and are focused on the objective of  representing CSE and associating them to standard language to translate encrypted messages. The ontology provides alternatives for representing weights and rules that determine the degree of suspicion. The solution uses it as an application ontology for selecting suspicious messages. Therefore, it is out of the scope of this ontology to represent the entire crime domain."    

- Listing 1: what do you mean by inference rules? Axioms that invoke OWL inferences? Rules expressed in another separate rule language (like SPARQL, SWRL)? Also please explain the rule syntax.

Response: In the new version, we clarify that SWRL was used in our ontology, and we provide a reference to SWRL standard: “The Semantic Web Rule Language (SWRL) [77] is used to express  knowledge-level inferences and queries. Listing 4.2.3 presents examples of SWRL rules defined to relate the acts described in the illustrative scenario to criminal activities.”  The rules in listing 1 are now in the standard SWRL (instead of simplified syntax). 

- Line 658: What doe "have weights associated individually" mean?

Response: The sentence was rephrased as follows:” ... have suspicious values associated individually to each term as specified in the ontology instances (e.g.,“Águia” - 1 and “big skull” - 4) ...”

- Line 703-704: How can a tokenization process translate phrases??

Response: The sentence was rephrased as follows: “This process translates key-phrases on the original messages using the OntoCexp, each sentence is transformed in a set of tokens, and the translation is processed token-by-token or grouped as key-phrases.

- It is not clear to me how Table 4 relates to Tables 1-3.

Response: Table 5 (Table 4 in the older version) includes the measures of Table 2 and 3 (Tabel 1 and 2 in the older version) but using 80-20 validation instead of cross-validation. We included the following sentence: “Table 5 presents the best results obtained from all 60 training/test executions performed using the dataset divided into two sets of 80% and 20% for training and testing, including the results by class and the overall value for F1-Score and Accuracy.”

- Line 871-874: Is this really what LEAs would want (see earlier comment)? And why is keyword search only against the original post? I would assume you would use the whole ontology in this case, to index the posts, i.e. both with translated words and slang?

Response: The users can select if (s)he will use the original posts or translated one, in another  similar search interface. As they are practically the same, we chose to show only one of them so as not to make the article even longer. From a technical and research point of view, we are working on query expansion mechanisms and visualization techniques to be used in this context, which will be the focus of a new research project.    

- Page 25: The first paragraph in the list presents what class it discusses, but none of the subsequent ones do, so it is not clear what they exemplify at all. Here would be a good place to also look at negative examples, issues, limitations etc.

Response: We included a  paragraph with the key limitations, with an example: “These examples illustrate the framework’s ability to retrieve and classify relevant posts of all studied ilocution classes. The main limitations of the framework is related to retrieving and classifying relevant posts with indirect speech acts [93], and those are context-dependent. For instance, the post “você deve levar mais açucar” (youmust takemore sugar – in English language – translated by the authors) apparently it is not a suspicious post, however information about the context (e.g., who spoke and situation) can turn it a suspicious one, since the term açucar (“sugar”) can be related to cocaine.”.  

- Line 921: not clear what multi-strategy refers to.

Response:   The sentence was rephrased as follows: Existing studies (e.g. [59,62,73]) proved that a strategy that combines techniques (e.g., ontologies and ML) is a more viable path to deal with our problem.

- Line 936: I am not convinced that they perform well.

Response: This statement has been removed.

- Line 940: interesting to know, but why did you not present this when you talked about the ontology before? What are the axioms? What are they used for? And how is the terminology expressed if the ontology only has classes, properties and a few axioms? Is it through annotations? Instances?

Response: In the new version, more details about the ontology and rules were included. The numbers have also been updated to the latest version of the ontology.

- Line 973: what balancing techniques? I did not see any.

Response: We used the SMOTE technique, a reference to this technique was included. 

- Reference list: there are some incomplete and some incorrect references, please review and revise the list. For instance, Lecture Notes in Computer Science is not a journal name, it is the name of a book series, where each book has an individual title, and it is this title that should appear in the reference, not only LNCS. Reference 85 has the wrong name of the proceedings volume, it is a workshop not a conference etc.

Response: References have been revised in the new version. 

Language issues:

Line 40: hug -> huge

Line 42: reformulate "turn difficult"

Line 421: reveled -> revealed?

Line 518: Coaine -> Cocaine

Line 666: in which extend -> to what extent

Line 864: set posts -> set of posts

Line 883: massages -> messages

Line 999: In this sence, does not fit, rephrase

Line 1001: select -> to select

Response: The mentioned language issues were corrected in the paper. The paper was fully revised with proof reads to improve readability. 

Reviewer 2 Report

The paper describes an approach for classifying tweets into their "intentions", according to the illocution framework proposed in [22]. The focus is on criminal messages. Authors use an ontology for rephasing slang terms, a weighting schema for determining whether the text is related to criminal content, and a classifier for classifying intentions. Overall, the method is interesting, but I think the paper needs some reorganization. Some parts are very verbose. For instance, the pictures of RNNs are unnecessary. The systematic review is interesting, but I think some parts can be replaced by graphs or tables rather than a sentence, so the reader can have a quick glimpse of the literature. Figure 4 adds little and Figures 5 and 6 are hard to follow from the text description.  A table summarizing the number of sentences in each class could be added. As no message from the Palinode class was tagged, I think this column should be removed from the results' tables (they are used for computing average statistics in table 4?), or some footnote mark should be added. It is not clear why some cells are colored into tables. Finally, some translations of twitter messages in the text are word-by-word. For the sake of clarity, I think that a translation of the meaning could be added.

Author Response

Dear reviewer, 

The reviewers’ comments on the first version of our manuscript were very helpful, and we believe our article improved thanks to them. 

Please, find below the response and changes for each comment. 

The paper describes an approach for classifying tweets into their "intentions", according to the illocution framework proposed in [22]. The focus is on criminal messages. Authors use an ontology for rephasing slang terms, a weighting schema for determining whether the text is related to criminal content, and a classifier for classifying intentions. Overall, the method is interesting, but I think the paper needs some reorganization. Some parts are very verbose. For instance, the pictures of RNNs are unnecessary. 

Response: We removed the neural network pictures, and we have summarized the presentation on neural networks to be more balanced with other techniques. Several parts of the paper were revised to be more concise and clear.  

The systematic review is interesting, but I think some parts can be replaced by graphs or tables rather than a sentence, so the reader can have a quick glimpse of the literature.

Response: We improved the review about the selected papers, as well as we highlight the differences of our proposal in relation to other works (Section 3). In SubSection 2.2, we included 6 references that approach the ontology-based social network analysis. We also updated links to source-code and the new version of the ontology.

Figure 4 adds little and Figures 5 and 6 are hard to follow from the text description.  

Response:  We improved the text description, particularly we clarified how we used the ontology in the framework. The overall presentation of the methods were improved, several descriptions were included and key aspects (e.g., the use of SWRL) were detailed. 

A table summarizing the number of sentences in each class could be added. 

Response:  We added a new table with these numbers (now Table 1). We rephrased the presentation of the class distribution of our training/test dataset and included a new table: ““Our training/test dataset includes (cf. Table 1):  267 inducements, 150 assertions, 116 proposals, 78 valuations, 54 wish, 21 forecasts, and 16 contrition, totalizing 704 tweets. No message (in our case each tweet has only one message) was classified as palinode.”   

As no message from the Palinode class was tagged, I think this column should be removed from the results' tables (they are used for computing average statistics in table 4?), or some footnote mark should be added. 

Response:  We changed the palinode class values to "-" and included a footnote explaining that this class was not used to compute the average score. 

It is not clear why some cells are colored into tables. 

Response: We provided explanations regarding the meaning of the colors in tables of results. We expect this can facilitate the readability of the results.  

Finally, some translations of twitter messages in the text are word-by-word. For the sake of clarity, I think that a translation of the meaning could be added.

Response: In the new version, we placed the word-by-word translation as footnote, and we added new translations in the text.  

Reviewer 3 Report

In this paper, authors presented ontology and ML-based system for the classification of social media text to detect criminal intention in posts. Authors presented each step of the proposed work clearly and deeply. I appreciate their research work. However, ontology has been used for the analysis of social media data, and authors missed to discuss those important works. Therefore, I recommend to discuss the following existing work in the manuscript that readers can understand the use of ontology in the domain of social network analysis. 1. Opinion mining based on fuzzy domain ontology and Support Vector Machine: A proposal to automate online review classification, 2. Fuzzy ontology-based sentiment analysis of transportation and city feature reviews for safe traveling, 3. Type-2 fuzzy ontology-based opinion mining and information extraction: A proposal to automate the hotel reservation system,4. Merged Ontology and SVM-Based Information Extraction and Recommendation System for Social Robots, 5. Transportation sentiment analysis using word embedding and ontology-based topic modeling, 6. A Fuzzy Ontology and SVM–Based Web Content Classification System. (include these in Section 2.2. Semantic web and Ontologies). This will help readers to know different research direction in the domain of ontology-based social network analysis.

The rules presented in listing 1 are confusing, authors should recheck these rules carefully. If these are SWRL rules than authors should clearly mentioned in the paper that these are SWRL rules.

Author Response

Dear reviewer, 

The reviewers’ comments on the first version of our manuscript were very helpful, and we believe our article improved thanks to them. 

Please, find below the response and changes for each comment. 

In this paper, authors presented ontology and ML-based system for the classification of social media text to detect criminal intention in posts. Authors presented each step of the proposed work clearly and deeply. I appreciate their research work. 

Response: We thank the reviewer for the compliments.

However, ontology has been used for the analysis of social media data, and authors missed to discuss those important works. Therefore, I recommend to discuss the following existing work in the manuscript that readers can understand the use of ontology in the domain of social network analysis. 1. Opinion mining based on fuzzy domain ontology and Support Vector Machine: A proposal to automate online review classification, 2. Fuzzy ontology-based sentiment analysis of transportation and city feature reviews for safe traveling, 3. Type-2 fuzzy ontology-based opinion mining and information extraction: A proposal to automate the hotel reservation system,4. Merged Ontology and SVM-Based Information Extraction and Recommendation System for Social Robots, 5. Transportation sentiment analysis using word embedding and ontology-based topic modeling, 6. A Fuzzy Ontology and SVM–Based Web Content Classification System. (include these in Section 2.2. Semantic web and Ontologies). This will help readers to know different research direction in the domain of ontology-based social network analysis.

Response: Section 2.2. (Semantic web and Ontologies) was improved based  on these recommendations. We discussed existing literature that uses ontologies or ontology-based frameworks in the domain of social network analysis. We included a summary of the suggested references. 

The rules presented in listing 1 are confusing, authors should recheck these rules carefully. If these are SWRL rules than authors should clearly mentioned in the paper that these are SWRL rules.

Response: The rules presented in listing 1 were revised and updated. The use of SWRL rules is now mentioned in the text. An introductory phrase was included, in addition to a reference to the document that contains the specification for a Semantic Web Rule Language (SWRL) from the W3C. The explanation of how the rules work has also been entirely revised (end of section 4.2.3). 

Round 2

Reviewer 1 Report

Thanks to the authors for the improvements of the paper, and for publishing the correct version of the ontology. Overall, the clarity of the paper has indeed improved quite a bit, and the role of the ontology has been made much more clear. Some choices of methods/technologies are still not motivated very well, and it is still not entirely clear how useful and usable the methods are in practice (e.g. depending on the relatively low accuracy, as well as the potentially high effort of updating the ontology). However, at least now these shortcomings are clearly evident from the paper and the reader can see what has actually been evaluated and what it left for future work, or others to determine.

I would recommend that the paper can be published with a few revisions, including:

Why are the new references 36-41 not considered related work? I would like to see a discussion on how this actually relates and compares to the approach proposed here. In the paper these papers are just briefly introduced, but there is no discussion on how they compare to the proposed solution in the paper.

I cannot make any sense of the two last sentences that were added to section 2.3, please rephrase/clarify. What is the ontology mentioned here?

Although the description of the ontology is much better now, the representation and usage of the weights is still a bit unclear. Could you provide an actual example, with weight properties, show how they are updated during configuration, and then used in the deployment phase. In particular, it is also still not clear why/how weights are manually updated when the ontology is configured - how is the developer able to determine such weights? Are they based on statistics of word usage/occurrence? If yes, then why do they need to be updated? If not, then what are they based on? In addition, on line 737 the authors say that weights are simply summed up, however, doesn't this have the side effect that longer Tweets will always be considered more suspicious?

Further, it is still not entirely clear why the SWRL rules are needed, and what they are used for. Why do you need to produce these new instances at runtime? Isn't it enough to just annotate the texts with existing terms in the ontology, rather than producing a set of new instances? It is also not clear how these rules are constructed. It seems almost impossible to construct rules for any possible combinations of terms that may occur in a Tweet, so how did you decide what rules are needed? And how they should be formulated?

It is still not entirely clear to me how the 8 million Tweets can be considered the evaluation dataset, rather than the 702. On lines 777-778 the authors state that the full set is used for evaluating "the other framework steps" - please be more clear exactly in what evaluations, i.e. what results presented in the paper, use the full set and what results were produced using the smaller set.

Finally, I do acknowledge that the combined approach seems to improve on the baseline, however, the results still seem quite modest, in terms of accuracy. I would appreciate a discussion in the paper on how useful a system with this level of accuracy could be. Considering that this is a quite hard task, and monitoring Tweets "by hand" is probably almost impossible, I assume that the authors can make a good argument that this is still useful.

Further language issues/typos/unclear sentences:

Line 149: "The relationship between four sets is capable of defining an ontology:" does not make sense, sets can not really be capable of anything - rephrase.

The acronym SVM is introduced on line 173, but is already used in the paragraph before. Please introduce acronym first time it is to be used.

Line 176: how can a blacklist provide accuracy and speed? Unclear.

Line 176-177: "ontology extracts..." - an ontology cannot extract anything, an ontology is a model. An ontology can be used within a system that extracts something.

Line 254: the Liu's --> Liu's

Line 470: do you really mean vulnerabilities? Against what threats?

Line 518: post' --> post ??

Line 521-522: do you mean that each term has a dataproperty who's value is a weight? Or does the property has a weight attached?

Line 665: What does "We used the false negative assignments to review OntoCexp." mean?

Line 668: thanks for publishing the new version of the ontology, however, it would be useful with a bit more documentation, e.g., such as comments inside the ontology.

Line 722: I think the correct English term would be "blow up" and not "explode"

Line 742: What does "revive a labels" mean?

Line 797: cross-revised --> cross-reviewed?

Line 1034: proved --> showed (since I am pretty sure they did not formally prove it)

There are still a few references where "Lecture Notes in Computer Science" is erroneously given as the book title of the volume. Please make another pass and double check the references.

Author Response

Dear reviewer, 

The reviewers’ comments on the second version of our manuscript were very helpful, and we believe our article improved thanks to them. 

Please, find below the response and changes for each comment. 

>>Thanks to the authors for the improvements of the paper, and for publishing the correct version of the ontology. Overall, the clarity of the paper has indeed improved quite a bit, and the role of the ontology has been made much more clear. Some choices of methods/technologies are still not motivated very well, and it is still not entirely clear how useful and usable the methods are in practice (e.g. depending on the relatively low accuracy, as well as the potentially high effort of updating the ontology). However, at least now these shortcomings are clearly evident from the paper and the reader can see what has actually been evaluated and what it left for future work, or others to determine.

I would recommend that the paper can be published with a few revisions, including:

Why are the new references 36-41 not considered related work? I would like to see a discussion on how this actually relates and compares to the approach proposed here. In the paper these papers are just briefly introduced, but there is no discussion on how they compare to the proposed solution in the paper.

> Response: 

Reviewer 3 suggested to include the proposed references in Subsection 2.2 in the first round of revision. We understand and accept the suggestion to further discuss it in Section 3 (Related Work). We included in Subsection 3.3 a discussion on how they compare to the proposed approach.  

>> I cannot make any sense of the two last sentences that were added to section 2.3, please rephrase/clarify. What is the ontology mentioned here?

> Response: These sentences were rephrased as follows: “Bonacin [46,47] proposed the Communication act Ontology (CacTO), which was designed to represent concepts of the pragmatic communication analysis. In this ontology, the communication acts are classified according to three dimensions (invention, time, and mode). Each communication act is linked to its performer and addressee. The message class includes Dataproperties (float values) to represent each illocution dimension and to describe the function part. The content is represented by links to external domain ontologies (using Objectproperties). Preliminary results showed the increase of the F1-Score from 0.77 (using only domain ontologies) to 0.86 (using CacTo integrated with domain ontologies) on scenarios of the special education domain [47]. A similar approach was explored on scenarios of electronic health records retrieval [48]. The F1-Score has increased from 0.65 to 0.76 by considering the illocution dimensions.”

>> Although the description of the ontology is much better now, the representation and usage of the weights is still a bit unclear. Could you provide an actual example, with weight properties, show how they are updated during configuration, and then used in the deployment phase. In particular, it is also still not clear why/how weights are manually updated when the ontology is configured - how is the developer able to determine such weights? Are they based on statistics of word usage/occurrence? If yes, then why do they need to be updated? If not, then what are they based on? In addition, on line 737 the authors say that weights are simply summed up, however, doesn't this have the side effect that longer Tweets will always be considered more suspicious?

> Response: We included additional explanations of how we adjusted the weights in our case study: “We adopted the following procedure to assign suspicious values. During the first iteration, each ontology CSE term (instances dataproperties) had their values inversely adjusted to their occurrence in the initial dataset. For instance, “açucar” (``sugar'' in English language) and “farinha” (“flour” in English language) are very common terms used to refer to drugs, but they are much more frequent in normal tweets in the analyzed dataset. These terms are in the first fifth of most frequent words in tweets among the terms represented in the ontology, so we attributed the weight 1 to these terms, whereas the crime specific terms received higher values.  For instance, “Vai de vala” (``go to the ditch'' in English language) is not frequent in normal (Portuguese language) tweets. This term received initially the weight 5, since they are among the 20% less frequent in the tweets. 

The iterative process was made up of adjustments according to the researchers' interpretation of the selected sentences (i.e., using the previously assigned values). When a non suspicious post was selected, the researchers analyzed the reasons and decreased the suspicious weight value of the terms linked to this post. For instance, the term “Águia” (``eagle''  -- in English language) is not frequent in the initial dataset of tweets and received a high suspicious value (5); after the analysis the researchers reduced the value to 1, since there were selected posts referring to eagle (animal). Weights were also assigned to the combinations of terms (modeled by the SWRL rules). For instance, when a post includes ``eagle'' and other terms (“e.g”,``let's explode'' and ``to hood'') this is a high suspicious post.”.

>> Further, it is still not entirely clear why the SWRL rules are needed, and what they are used for. Why do you need to produce these new instances at runtime? Isn't it enough to just annotate the texts with existing terms in the ontology, rather than producing a set of new instances? It is also not clear how these rules are constructed. It seems almost impossible to construct rules for any possible combinations of terms that may occur in a Tweet, so how did you decide what rules are needed? And how they should be formulated?

 >Response:  We included additional explanations about the modeling and use of the SWRL rules: “Some ontology design decisions were made when choosing the use of SWRL, as well as we defined a procedure to model and use the rules. The main reason we chose to use SWRL is due to its high-level abstract syntax for Horn-like rules. We consider this syntax adequate to represent how combinations of concepts, present in posts, can be evaluated (in the rule body/antecedent) to infer criminal acts suspicious (in the rule head/consequent). Complex rules can be defined by using SWRL, including chained inferences about new facts, preparing the framework for further improvements, even if the current version uses simpler and more direct rules.     

In the framework, these rules can be fired using the following procedure: (1) new acts (instances of the Act class - Act(?ac01) are created when a post is analyzed by the framework (using Protégé API); (2) the framework reads the post and it links (using objectProperties) this act according to the terms used in the post (using Protégé API); (3) a reasoner\footnote{http://www.hermit-reasoner.com/} evaluates the rules and, if the new action matches with its body, it associates the act with criminal actions and respective weights; and, finally, (4) the framework reads the weights and attributes a suspicious level to the post. In the current version, the new instances (from the analyzed posts) can be saved for debugging and storage proposals. 

The current set of rules was created from the evaluation of two researchers (co-authors) of the terms used in the ontology. Although it is impracticable to create rules for all possible combinations, this is an extendable model that can evolve into an increasingly representative set over time, through the analysis of new tweets and the concepts of the ontology.”

>> It is still not entirely clear to me how the 8 million Tweets can be considered the evaluation dataset, rather than the 702. On lines 777-778 the authors state that the full set is used for evaluating "the other framework steps" - please be more clear exactly in what evaluations, i.e. what results presented in the paper, use the full set and what results were produced using the smaller set.

> Response:   We rewrite these sentences as follows:  ”The entire set of tweets (i.e., 8,835,016  tweets) was used to evaluate the steps A1, A2 and A3 of the OFCIC Training Component (Figure 3) and steps B1, B2 and B3 of the OFCIC Intention Classification Component (Figure 4). The 704 tweets were used to train and test the ML model (items A4, A5, A6, A7 and A8 of Figure 3) and to evaluate the results (items B4, B5, B6, B7 and B8 of Figure 4). Therefore, ML results refer to cross validation (and 80-20 split) using 704 tweets.”.

>> Finally, I do acknowledge that the combined approach seems to improve on the baseline, however, the results still seem quite modest, in terms of accuracy. I would appreciate a discussion in the paper on how useful a system with this level of accuracy could be. Considering that this is a quite hard task, and monitoring Tweets "by hand" is probably almost impossible, I assume that the authors can make a good argument that this is still useful.

> Response: We included additional explanations about this aspect at the discussion section: “From a practical point of view, one of the challenges is to increase the obtained results from the ML techniques. The overall result around 0.5 is relatively low, and even individual class results (it may be what matters in practice and are higher, as discussed) may not be the expected for law enforcement agencies. However, the illocution classification is an additional tool to provide filtering and ranking options for suspicious messages selected by the framework. Despite the current results, these tools can be useful in practice when we consider the huge number of social network posts, which makes manual assessment practically impossible. Besides, investigators may be frequently interested only in highly suspicious posts, which are in the first positions of the rank and are more likely to be recovered by the framework, due to the occurrence of CSE.”

>> Further language issues/typos/unclear sentences:

Line 149: "The relationship between four sets is capable of defining an ontology:" does not make sense, sets can not really be capable of anything - rephrase. 

The acronym SVM is introduced on line 173, but is already used in the paragraph before. Please introduce acronym first time it is to be used.

Line 176: how can a blacklist provide accuracy and speed? Unclear.

Line 176-177: "ontology extracts..." - an ontology cannot extract anything, an ontology is a model. An ontology can be used within a system that extracts something.

Line 254: the Liu's --> Liu's

Line 470: do you really mean vulnerabilities? Against what threats?

Line 518: post' --> post ??

Line 521-522: do you mean that each term has a dataproperty who's value is a weight? Or does the property has a weight attached?

Line 665: What does "We used the false negative assignments to review OntoCexp." mean?

Line 668: thanks for publishing the new version of the ontology, however, it would be useful with a bit more documentation, e.g., such as comments inside the ontology.

Line 722: I think the correct English term would be "blow up" and not "explode"

Line 742: What does "revive a labels" mean?

Line 797: cross-revised --> cross-reviewed?

Line 1034: proved --> showed (since I am pretty sure they did not formally prove it)

There are still a few references where "Lecture Notes in Computer Science" is erroneously given as the book title of the volume. Please make another pass and double check the references.

> Response: These small corrections were addressed in the paper. We are also working on the documentation of the ontology.

Reviewer 2 Report

Authors have addressed my main concerns, and I have no further comments.

Author Response

We would like to thank you for the valuable comments on the first review round.